# The AAA + ATPase TorsinA polymerizes into hollow helical tubes with 8.5 subunits per turn

F. Esra Demircioglu [1], Weili Zheng [2], Alexander J. McQuown[3], Nolan K. Maier[1,4], Nicki Watson[5], Iain M. Cheeseman [1,4], Vladimir Denic[3], Edward H. Egelman [2] & Thomas U. Schwartz [1]

TorsinA is an ER-resident AAA + ATPase, whose deletion of glutamate E303 results in the genetic neuromuscular disease primary dystonia. TorsinA is an unusual AAA + ATPase that needs an external activator. Also, it likely does not thread a peptide substrate through a narrow central channel, in contrast to its closest structural homologs. Here, we examined the oligomerization of TorsinA to get closer to a molecular understanding of its still enigmatic function. We observe TorsinA to form helical filaments, which we analyzed by cryo-electron microscopy using helical reconstruction. The 4.4 Å structure reveals long hollow tubes with a helical periodicity of 8.5 subunits per turn, and an inner channel of ~4 nm diameter. We further show that the protein is able to induce tubulation of membranes in vitro, an observation that may reflect an entirely new characteristic of AAA + ATPases. We discuss the implications of these observations for TorsinA function.

[1] Department of Biology, Massachusetts Institute of Technology, Cambridge, MA 02139, USA. [2] Department of Biochemistry and Molecular Genetics, University of Virginia, Charlottesville, VA 22908, USA. [3] Department of Molecular and Cellular Biology, Harvard University, Cambridge, MA 02138, USA. [4] Whitehead Institute for Biomedical Research, , Cambridge, MA 02142, USA. [5] W. M. Keck Microscopy Facility, The Whitehead Institute, Cambridge, MA 02142, USA. Correspondence and requests for materials should be addressed to T.U.S. (email: tus@mit.edu)

Torsins are essential proteins that belong to the AAA+ (ATPases associated with a variety of cellular activities) superfamily. AAA + ATPases encompass diverse enzymes that use ATP hydrolysis to drive protein and nucleic acid remodeling, protein degradation, and other functions[1–4]. They share a bilobed core of an N-terminal, 200–250 aa nucleotide-binding domain, and a C-terminal, ~50–80 aa small domain. The nucleotide-binding site has several characteristic signature motifs, including P-loop, Walker-A and -B, and sensors-1 and −2[5]. Typically, AAA + ATPases form hexameric ring or double-ring structures, in which neighboring subunits act as activators for ATP hydrolysis involving a highly conserved arginine residue ("Arg finger")[6]. The small C-terminal domain is critical for ring formation. Substrate engagement often involves threading through the narrow central channel of the ring structure[7,8]. Torsins are restricted to multicellular eukaryotes and they exclusively reside in the endoplasmic reticulum (ER) and the connected perinuclear space (PNS)[9,10]. Humans have four Torsins, with different tissue-specific expression profiles[11]. From a medical perspective TorsinA is important, since a deletion of glutamate 303 in TorsinA, TorsinAΔE for short, is the founding mutation for early-onset primary dystonia[12,13], a devastating and still incurable neuromuscular movement disorder[14–16].

Torsins are intriguing since they differ from their closest structural homologs in surprising ways. Among AAA+ ATPases, they are, sequence-wise, most similar to bacterial Clp proteins, well-understood molecules engaged in protein unfolding and degradation[17]. However, the similarity to Clp proteins proved to be rather misleading with regard to the search for the elusive Torsin function. Our previous analysis indicated that torsins lack pore loop consensus motifs, which Clp proteins use for threading unfolded protein substrates through their central channel[18]. In addition, the two tightly interacting proteins Lamina-Associated Protein 1 (LAP1) and LUminal Domain Like LAP1 (LULL1) were initially considered to be substrates, since they preferentially bind to ATP-bound TorsinA, similar to Clp substrates[19–21]. However, we now know that LAP1 and LULL1 are activators of TorsinA, with a curious structural similarity to AAA + ATPases[22–24]. While LAP1 and LULL1 provide the Arg finger for activating ATP hydrolysis, they are unable to bind nucleotide themselves, due to a lack of the characteristic sequence motifs introduced above[24]. The dystonia mutant TorsinAΔE weakens the interaction with the activators, which makes it a loss-of-function protein[18,20–22]. LAP1 and LULL1 are both transmembrane proteins[19,25], which raises the possibility that TorsinA has a novel membrane-associated function. In addition, Torsins are different from other AAA + ATPases because of a set of highly conserved cysteine residues, positioned in a way that suggests them to be coupled to ATP hydrolysis, and potentially involved in a redox mechanism[18,21,26]. Furthermore, Torsins contain a hydrophobic N-terminal region that likely plays a role in membrane binding[27,28].

The structural studies on TorsinA and its activators provoked another question, namely the oligomerization state of TorsinA. Based on the similarity of TorsinA to well-known AAA + ATPases, a heterohexameric ring assembly of three TorsinA subunits alternating with three activator subunits initially made logical sense[23,24]. In addition, there was circumstantial evidence for this conclusion, such as stoichiometric and low-resolution structural data[23,24]. However, the activators LAP1 and LULL1 lack the small C-terminal domain, and their surfaces are not well conserved in the areas that would be expected to interact with TorsinA to form oligomers beyond the catalytically activatable heterodimer[18], putting a heterohexameric ring assembly in doubt (see discussion in refs. [18,29]). On the other hand, various publications presented data that suggested oligomerization of

TorsinA alone, albeit at limited resolution[11,29–31]. In this study, we probed the oligomeric state of TorsinA in solution. We observed that TorsinA, presumably when ATP-bound, can readily assemble into continuous helical filaments. These structures, again, are different from canonical AAA + ATPases, since we observe a periodicity of 8.5 subunits per turn rather than six, and a much larger central channel. Our in vitro experiments performed with liposomes suggest that the lining of the inner channel of these filaments may bind directly to lipids, likely relevant to elucidating the enigmatic function of TorsinA.

## Results

**Self-assembly of TorsinA.** To examine the homo-oligomerization of TorsinA, we designed a series of constructs to recombinantly express TorsinA in bacteria with different solubility tags. An N-terminally MBP-tagged human TorsinA construct (residues 51–332) was produced in high yield and purity (Supplementary Fig. 1a, b). We obtained large amounts of MBP-TorsinA through a one-step affinity purification at a purity greater than 95%. Subsequent gel filtration analysis revealed that the majority of the protein eluted in the void volume of the column, indicating a size of over 600 kDa, thus higher-order oligomerization (Supplementary Fig. 1c).

To understand the exact nature of these MBP-TorsinA assemblies, we first performed a negative-stain analysis of the affinity-purified protein. The sample grids showed TorsinA assembled into filamentous structures with an obvious internal order, suggesting that helical reconstruction may be a feasible approach for structure determination (Supplementary Fig. 1d). Next, we explored options to obtain even longer filaments. Proteolytically cleaving MBP with 3C protease, combined with ATP-containing buffer resulted in longer filaments, which were then suitable for helical reconstruction.

**Structural analysis of TorsinA filaments.** The longer TorsinA filaments obtained after 3C cleavage, clustered into thick bundles on negatively stained EM grids at low-ionic-strength buffers, whereas at high-ionic-strength conditions they remained separated (Supplementary Fig. 1e, f). We obtained the most suitable sample for structural analysis at a buffer condition containing 300 mM NaCl. We next prepared frozen-hydrated cryo-EM grids in this improved buffer condition to helically reconstruct TorsinA filaments at higher resolution. We were able to collect useful images from the grid areas containing very thin ice (Fig. 1a). Using 75,909 overlapping segments from 602 micrographs, we helically reconstructed TorsinA at ~4.4 Å resolution (Fig. 1; Supplementary Figs 2, 3, and Table 1). The resulting structure showed TorsinA to assemble into a helical filament with an outer diameter of ~14.5 nm. Surprisingly, and in contrast to most AAA + ATPases, we observe that the filament has a central channel with a diameter of ~4 nm. The filaments have a right-handed 1-start helix with a pitch of ~47 Å, and they contain 8.5 TorsinA molecules per turn with an axial rise of 5.5 Å per subunit (Fig. 1b–d). Non-hexameric assemblies of AAA + ATPases, while rare, have been observed before, particularly in DNA-remodeling enzymes (see Discussion below). The EM density for the small domain of TorsinA is less well defined than the large domain, potentially indicating some flexibility. Also, each of the helical turns in the TorsinA filament is in close contact with a neighboring turn (Fig. 1d; Supplementary Fig. 4b). Interestingly, the disease-causing ΔE mutation lies in the proximity of these contact sites and the ΔE mutant did not polymerize as judged by negatively stained EM grids (Supplementary Figs 4b, 5f).

The superposition of filamentous TorsinA with its hetero-dimeric LULL1-bound state[18] shows that the protein rearranges

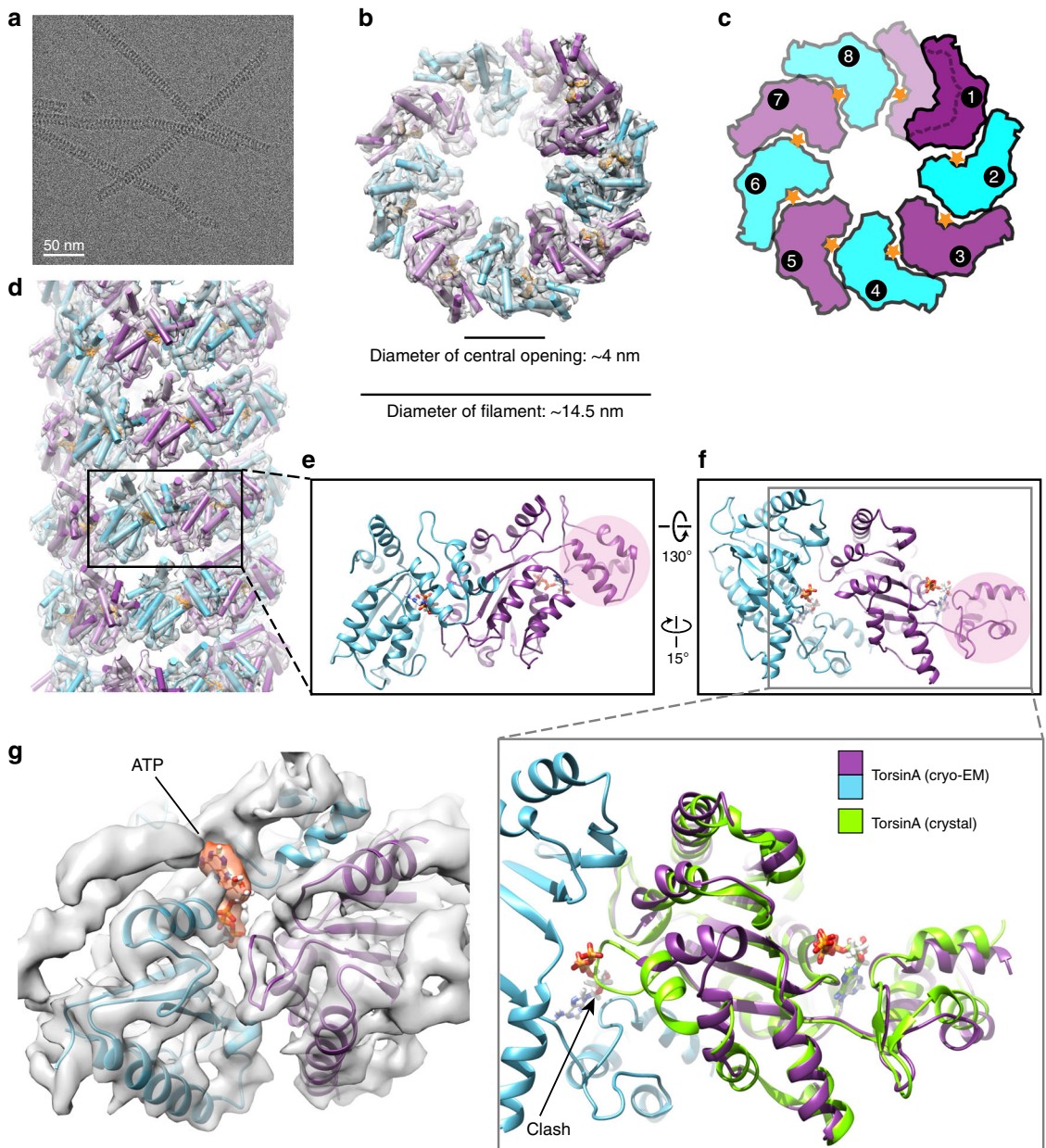

**Fig. 1** Cryo-EM reconstruction of the TorsinA filaments. **a** A cryoelectron micrograph of the TorsinA filaments embedded in thin ice. **b** End view of the reconstructed helical tube fitted into the cryo-EM density where TorsinA subunits are shown in alternating colors of purple and cyan, and ATP is shown in orange. **c** Schematic drawing of the TorsinA filament colored as in (**b**). Orange stars represent ATP molecules. A single helical turn contains 8.5 TorsinA subunits. **d** Side view of the reconstructed TorsinA tube depicted and colored as in (**b**). Cartoon representation of a TorsinA–TorsinA dimer element from within the helical tube is shown in two different orientations in (**e**) and (**f**). The small domain of a single TorsinA in the dimer is highlighted in pink. The previously determined crystal structure of TorsinA (PDB: 5j1s) is superposed on the TorsinA–TorsinA dimer in an enlarged view of (**f**). The non-catalytic face of TorsinA rearranges in the cryo-EM structure to avoid a steric clash with the nucleotide. **g** Nucleotide-binding region of the TorsinA–TorsinA dimer fit into the cryo-EM density. Bound nucleotide is modeled as ATP and highlighted in orange color in the density

significantly (Fig. 1e, f; Supplementary Fig. 6). The conformation of TorsinA in the LULL1-bound state is incompatible with filament formation due to steric hindrance. The biggest change occurs at the non-catalytic face of TorsinA. There, the rearrangement of residues 232–262, is critical to prevent steric clashes with the nucleotide bound to the neighboring unit (Fig. 1f; Supplementary Figs 4a, 6). A highly conserved glycine (G251) in this region is likely to be pivotal for the structural rearrangement to occur. To prove this point, we mutated this glycine to tyrosine (G251Y). As expected, the mutant protein did not polymerize, as

judged by negative-stain EM (Supplementary Fig. 5c, g, h). Notably, residue G251 has been mutated also in a previous study[29], disrupting TorsinA's oligomerization behavior in vitro and its physiological functioning in vivo. Helix α0, the loops following the helix α3, and the rearranged region reside at the subunit-subunit interface of the TorsinA helical tube (Supplementary Figs 4a, 6). Mutating the conserved residues D188 and D264 (D188A/Y, D264A) in this region also prevents filament formation, as observed by negative-stain EM (Supplementary Fig. 5a, b, d, g, h).

**Table 1 Cryo-EM data collection, refinement, and validation statistics**

|  | TorsinA (EMDB-20076) (PDB 6OIF) |
|---|---|
| *Data collection and processing* | |
| Magnification | 36,000 |
| Voltage (kV) | 200 |
| Electron exposure (e⁻/Å²) | 30 |
| Defocus range (μm) | 0.5–1.5 |
| Pixel size (Å) | 1.169 |
| Symmetry imposed | Rise: 5.5 Å; twist: 42.5° |
| Initial particle images (no.) | 75,909 |
| Final particle images (no.) | 69,670 |
| Map resolution (Å) | 4.4 |
| FSC threshold | 0.143 |
| *Refinement* | |
| Initial model used (PDB code) | 5J1S |
| Map sharpening $B$ factor (Å²) | −150 |
| Model composition | |
| Non-hydrogen atoms | 58,324 |
| Protein residues | 7050 |
| Ligands | ATP |
| r.m.s. deviations | |
| Bond lengths (Å) | 0.006 |
| Bond angles (°) | 1.391 |
| Validation | |
| MolProbity score | 1.93 |
| Clashscore | 6.18 |
| Poor rotamers (%) | 0 |
| Ramachandran plot | |
| Favored (%) | 88.56 |
| Allowed (%) | 11.44 |
| Disallowed (%) | 0 |

**Filamentous TorsinA is likely ATP bound**. The resolution of our helical reconstruction is not high enough to directly determine the nature of the bound nucleotide. We presume that the TorsinA filaments are nucleotide bound for two reasons: first, we observed EM density at the nucleotide-binding site (Fig. 1g), and second, we observed robust filament formation only in ATP-containing buffer. We also examined whether TorsinA filaments are catalytically active. We performed an NADH-coupled ATPase assay and observed no ATPase activity (Supplementary Fig. 7). This experiment suggests that it is the lack of an Arg finger in TorsinA that prevents ATP hydrolysis. To directly prove that ATP is present in the nucleotide-binding pocket, we also purified an ATP-trap mutant, MBP-TorsinA E171Q (residues 51–332), which should allow stable retention of ATP. Intriguingly, we did not observe this mutant forming filaments on negative-stain EM grids (Supplementary Fig. 5e). Considering this observation, and the flexible state of the small domain/sensor-2 motif, we conclude that ATP is likely bound in a noncanonical state in the TorsinA helical assembly. In the E171Q mutant, the ATP molecule may not be able to adopt this conformation.

**Membrane tubule formation by TorsinA**. TorsinA helical tubes have a wide inner channel, with an approximate diameter of ~4 nm. The electrostatic surface along the inner channel of TorsinA shows an undulating pattern of positively and negatively charged stripes (Fig. 2a–d). Most of the positively charged residues inside the channel are well conserved (Fig. 2e). Based on numerous publications that show TorsinA's potential role in membrane remodeling[13,32–35], we asked whether TorsinA may engage with phospholipid membranes through its inner channel. Such behavior would be reminiscent of dynamin- or BAR domain-like membrane remodelers[36–38].

To test this hypothesis, we prepared small acidic liposomes and incubated TorsinA with them under a variety of conditions. We either incubated liposomes, MBP-tagged TorsinA and 3C protease for about 6 h, long enough to cleave off about 50% of the MBP fusion tags (as judged by SDS-PAGE analysis) or, alternatively, we pre-cleaved the MBP tags overnight (about 80% cleavage efficiency) and mixed the long TorsinA filaments with liposomes for about an hour. In addition, we tested the uncleaved MBP-tagged TorsinA in our liposome assays. We also performed these experiments at different ionic-strength conditions to evaluate the effect of electrostatic interactions between TorsinA and lipids. Subsequently, we examined all our samples on negatively stained EM grids. At relatively low-ionic-strength conditions (80–100 mM NaCl), we observed numerous protrusions extending from the liposomes, regardless of whether the MBP fusion tags were cleaved or not (Fig. 3a–f; Supplementary Fig. 8a–c). Since we did not observe these protrusions in a higher ionic-strength buffer, TorsinA likely mediated their formation by acting electrostatically on lipids. The protrusions appeared to be decorated with a coat of TorsinA molecules, which suggests that membrane extrusion happened through the inner channel. However, we noted that these TorsinA-coated membrane tubules were noticeably larger in diameter (mean: 20 nm) than the membrane-free TorsinA filaments (mean: 15.5 nm) (Fig. 3g). The larger diameter suggests that filaments were arranged differently, presumably with more TorsinA molecules per helical turn, in order to establish a wider inner channel. For MBP-TorsinA, the difference in diameter between filaments and protrusions was less pronounced (Supplementary Fig. 8d). In addition, we noticed that TorsinA did not only coat the protrusions, but rather the entire membrane surface (Fig. 3c–f).

In order to understand how the lining of the inner channel of TorsinA may affect polymerization as well as the formation of membrane tubules, we introduced mutations on three highly conserved lysine residues (K148, K174, K184) and histidine H140 of the inner channel (Supplementary Fig. 9, Table 2). H140N, K174A, and K174E mutants all form membrane-free filaments. While H140N and K174E did not tubulate membranes, K174A did. In contrast, K148E and K184E mutants did not form membrane-free helical filaments. However, while the K184E mutant tubulated liposomes similar to wild-type TorsinA, the K148E mutant did not. These observations suggest that filament formation and membrane tubulation are not directly linked and are, according to our data, at least partially independent of each other.

Finally, we tested whether TorsinA hydrolyzes ATP while remodeling liposomes, different from the enzymatically inactive TorsinA filaments. To test this, we performed an NADH-coupled ATPase assay with our liposome-TorsinA mixtures (Supplementary Fig. 7). Again, we did not observe ATPase activity. Finally, the Walker B mutant (E171Q) of TorsinA also triggers membrane tubulation on liposomes (Supplementary Fig. 9g; Table 2), while it cannot form membrane-free filaments. This observation further supports the notion that the two helical arrangements are substantially different.

**TorsinA function at the inner nuclear membrane**. The hallmark of Torsin loss and/or loss-of-function in cells is nuclear blebs arising from the inner nuclear membrane (INM)[13,33,34,39,40]. Since the neck of these blebs is occupied by nuclear pore complex (NPC)-like structures, it is reasonable to speculate that TorsinA is involved in NPC assembly[35]. We asked whether a membrane remodeling activity of TorsinA as observed in our in vitro liposome assays contributes to the blebbing phenotype. For this, we first generated a HeLa cell line containing a triple knockout of

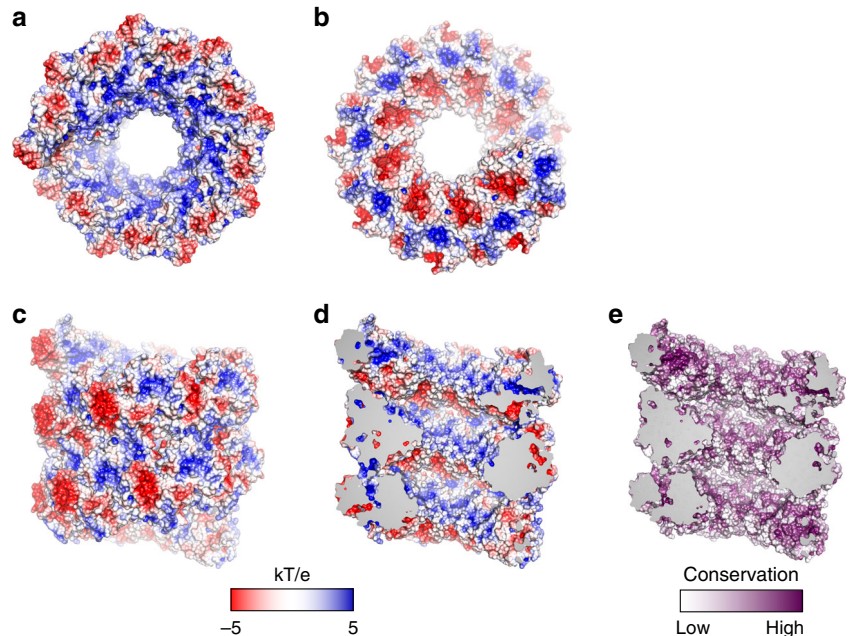

**Fig. 2** Electrostatic surface potential and conservation analysis of the TorsinA filament. **a**, **b** Top- and bottom-end electrostatic surface views of the TorsinA filament, revealing a basic character in (**a**) and an acidic character in (**b**) in tube interior. **c** Exterior electrostatic surface view of the TorsinA filaments. **d** A cutaway electrostatic surface view of the TorsinA helical tube, revealing an undulating pattern of positive and negative charges on the interior surface of the helical tube. **e** Surface conservation of TorsinA. Same view as (**d**). Basic residues on the interior surface of the helical tube are mostly conserved

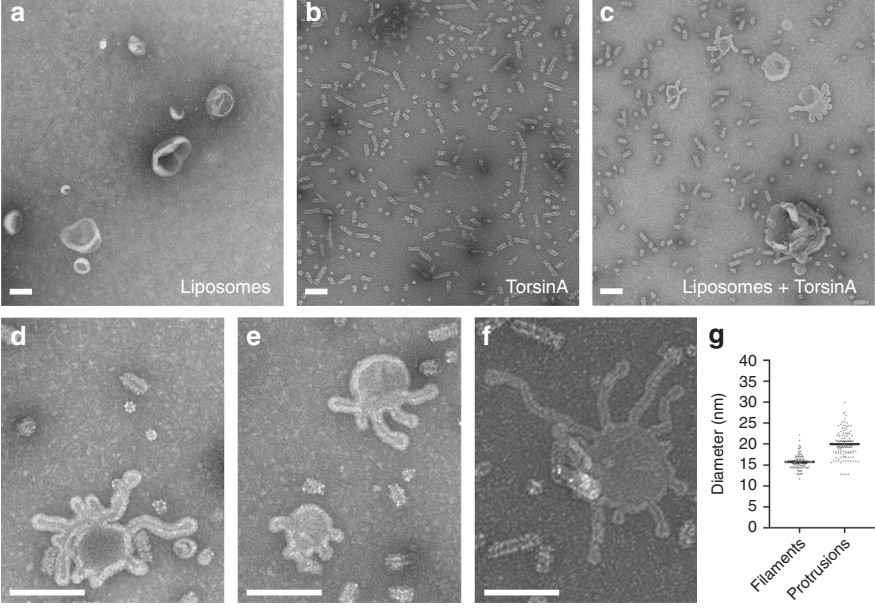

**Fig. 3** TorsinA tubulates acidic membranes in vitro. **a** Negative-stain electron micrograph of moderately acidic liposomes. **b** Negative-stain electron micrograph of MBP-TorsinA incubated with 3C protease for 6 h at RT. **c** Negative-stain image of the moderately acidic liposomes which are tubulated after incubation with MBP-TorsinA and 3C protease for 6 h at RT. **d**, **e**, **f** Higher magnification views of tubulated liposomes with protrusions of various diameter decorated with a coat of TorsinA. Scale bar is 100 nm in all micrographs. **g** Scatter plot of the diameters measured from negative-stain micrographs using ~100 filaments and ~100 lipid protrusions. Mean values are shown together with 95% confidence intervals

TorsinA/TorsinB/Torsin3A using CRISPR/Cas9 engineering (Supplementary Fig. 10a). Ultrastructural characterization of the sections obtained from these cells revealed a robust and pronounced blebbing phenotype at the INM (Fig. 4a, b, g). Next, we stably expressed TorsinA constructs in the triple-KO TorsinA/TorsinB/Torsin3A cells at near endogenous levels (Supplementary Fig. 10b). We expected that introducing the wild-type TorsinA would rescue the blebbing phenotype, while introducing the TorsinA mutants would not if the mutations were to interfere with a blebbing-relevant function. Since H140N, K148E, and K174E mutations impeded the ability of TorsinA to tubulate the liposomes in vitro, we expressed these mutants in our triple KO cells to examine their effects. Intriguingly, we observed that the wild-type along with all of the mutant TorsinA constructs rescued nuclear blebbing (Fig. 4c–g; Table 2). Thus, our data suggest that the membrane remodeling of TorsinA is either not directly linked

**Table 2 Biochemical and functional analysis of TorsinA variants**

|  | Filament formation in solution | Membrane tabulation on liposomes | Rescue of nuclear blebs |
|---|---|---|---|
| Wild-type | + | + | + |
| E171Q | − | + | Not tested |
| H140N | + | − | + |
| K148E | − | − | + |
| K174A | + | + | Not tested |
| K174E | + | − | + |
| K184E | − | + | Not tested |

to the blebbing phenomenon or that the in vivo experiment is less sensitive than the liposome assay.

## Discussion

This study was motivated by the uncertainty about the oligomeric state of TorsinA, which is an important impediment toward elucidating the biological function of TorsinA. We show, unambiguously, that TorsinA forms long helical filaments in solution, with a periodicity of 8.5 subunits per turn and an inner channel of ~4 nm diameter. How unusual is this for AAA + ATPases? While most AAA + ATPases form hexameric ring structures, TorsinA is not the only exception. A prominent, well-studied example is the MCM2–7 complex, the helicase involved in DNA replication[41,42]. In eukaryotes, the six different subunits form a functional head-to-head double-hexameric ring assembly[43]. The homologous helicase in archaea, however, only has one distinct subunit and has been shown to assemble into oligomers and polymers. For example, it can form helices with a periodicity of 7.2 subunits per turn, and it can also form eight-membered rings[44,45]. In both assemblies, a wide central channel of ~3–4 nm is observed, similar to what we observed in TorsinA filaments. Widening of the central channel is a direct consequence of having more subunits per turn. Comparing TorsinA to MCM is somewhat problematic, since MCM has a large N-terminal domain (NTD), absent in TorsinA. This NTD is directly involved in oligomerization, so the assembly structures of TorsinA and MCM only superpose generally. Another example for a non-hexameric AAA + ATPase assembly is the structure of DnaA, also a component of the replication machinery. Here, helical filaments with an $8_1$ symmetry were observed in a crystal lattice[46]. These DnaA helices are much more elongated with a rise of 17.8 nm per turn, compared to 4.7 nm in TorsinA. Crystal lattices can enforce symmetries, and they need to be carefully examined for physiological relevance. In the DnaA case, the authors argued that the DnaA interfaces were quite similar to established hexameric units, and that they had a nucleotide bound. Therefore, these assemblies should be physiologically relevant[46]. Using the same argument, we can make a strong case for the TorsinA filaments being biologically relevant, since the interfaces are similar to the established TorsinA–LULL1 interface, albeit with modifications in the 232–262 region. In addition, the filaments presented here are formed in solution rather than in crystals, so we can ignore packing forces associated with the latter.

One important, possible limitation of our study is that we used an N-terminally truncated TorsinA construct in our structural analysis. Could the structures therefore be an artifact? The most direct test would be to do the same study with the full-length TorsinA. Unfortunately, this protein is ill-behaved in our hands and it aggregates, thus preventing us from doing meaningful experiments. From a steric perspective, we cannot see a

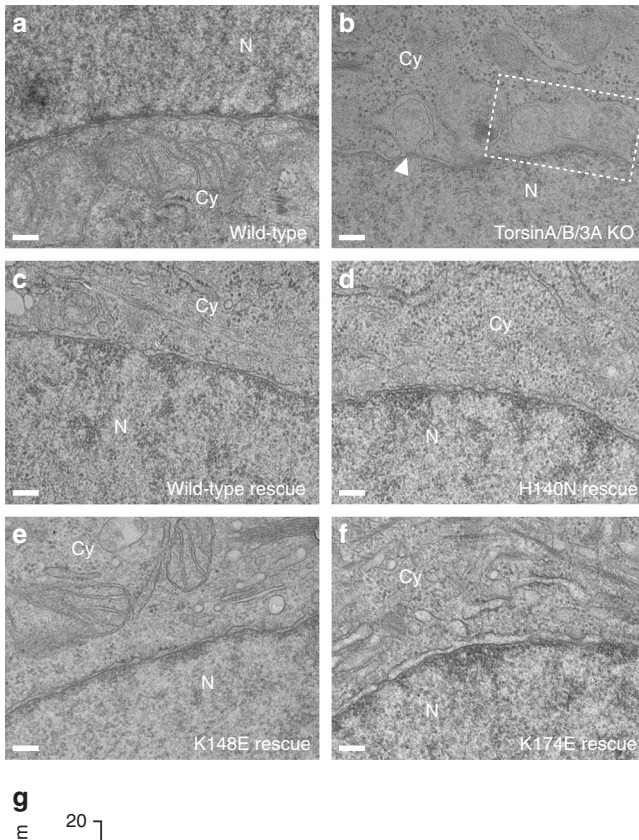

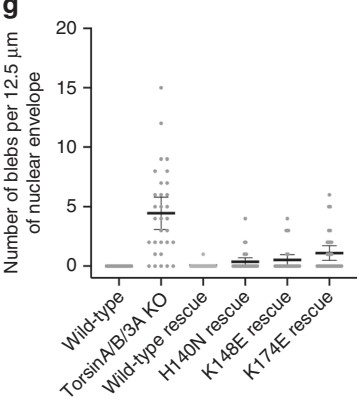

**Fig. 4** TorsinA mutants and the blebbing at the nuclear envelope. Representative EM cross-sections of (**a**) a wild-type HeLa cell, (**b**) a triple TorA/B/3A knockout HeLa cell, and (**c–f**) the triple TorA/B/3 A knockout HeLa cells rescued by stable expression of the TorsinA variants. Dashed box area encloses multiple blebs, and the white arrowhead marks the electron density observed at the curvature of the bleb neck. N nucleus, Cy cytoplasm. Scale bar, 150 nm. **g** Scatter plot of the number of blebs observed per EM cross-section (average NE membrane in each cross-section: 12.5 μm) in HeLa cell lines. Thirty EM cross-sections per cell line were analyzed. Mean values are shown together with 95% confidence intervals

convincing argument why the full-length protein should not be able to also generate filaments. The missing 30 N-terminal residues (aa 21–50) are presumably flexible, and can likely adopt various positions on the TorsinA surface. Thus, they should not block assembly.

Are helical filaments the only self-assembled form of TorsinA? If this were true, one would argue that the interface between adjacent subunits should be distinct from the canonical hexameric interface of other AAA + ATPases. We compared the TorsinA assembly with established hexamers of related AAA +

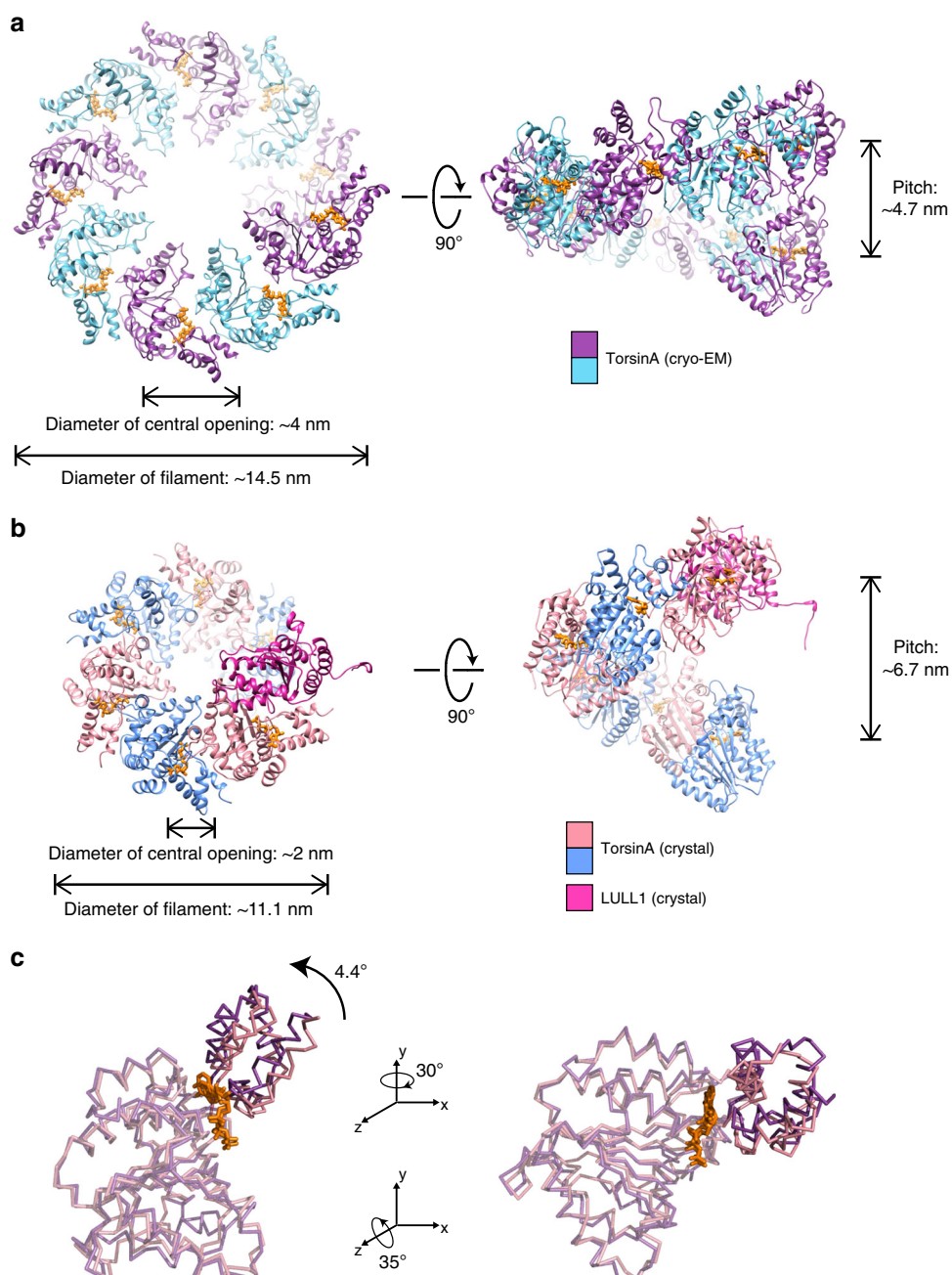

**Fig. 5** Making TorsinA filaments with different diameters and symmetries. End and side views of a single helical turn extracted from (**a**) the TorsinA filaments reconstructed by cryo-EM, (**b**) a modeled TorsinA filament based on the previously determined TorsinA–LULL1 crystal structure (PDB: 5j1s). The filament in (**b**) was obtained by repetitive superposition of TorsinA on LULL1, and thus the helical symmetry of the modeled helical tube was dictated by the structural arrangement of TorsinA and LULL1 in the crystal structure. **c** Ribbon representations of the TorsinA crystal and cryo-EM structures superposed on their large domains. Small domain is displaced through a 4.4° rotation between the two structures. Filaments with a range of diameters and symmetries may form upon minor changes in the TorsinA structure and/or small shifts in the structural arrangement between the adjacent Torsins

ATPases. The difference between the two assemblies is quite small, and only requires minor rotational adjustment of the large versus the small domain. In the TorsinA structure, we do not recognize any detail that would force a specific relative position of small versus large domain (Fig. 5). Therefore, we argue that TorsinA may adopt multiple oligomeric states in the ER. This is in full agreement with previous publications, where TorsinA was shown to form hexameric and other higher-order assemblies[11,29–31]. In a recent article, the higher-order assembly of Torsins was discussed in the context of its biological function[29]. The notion of various oligomeric and polymeric states for

TorsinA is also supported by our liposome interaction studies. We show that liposomes incubated with TorsinA undergo remodeling (Fig. 3). TorsinA triggers the formation of long protrusions and is able to coat the entire liposome, under the conditions tested. Presumably, the long protrusions are TorsinA coated, however, the diameter of these protrusions is variable and often wider than that of unbound TorsinA filaments. Unfortunately, the tubular protein coats assembled on liposomes are not amenable to helical reconstruction, so we cannot assess their helical parameters. However, the mean diameter of the protrusions is about 20 nm, compared with 15.5 nm for the

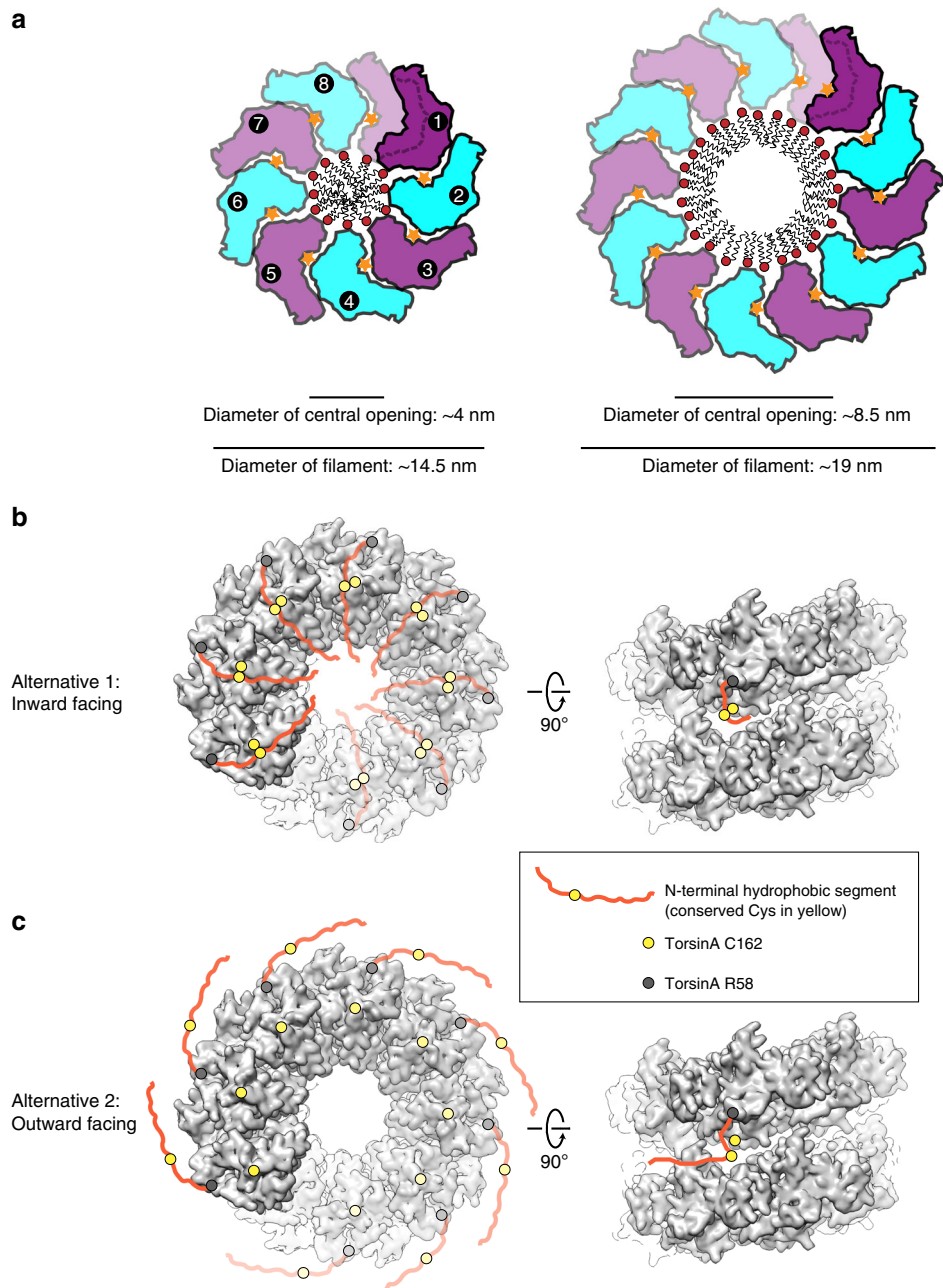

**Fig. 6** Model for the functioning of TorsinA filaments. **a** Schematic drawings of TorsinA filaments illustrate how TorsinA may engage the membrane in liposome protrusions. The left panel displays the filaments reconstructed by cryo-EM, whereas the right panel is modeled based on the average diameter of TorsinA-coated liposome protrusions shown in Fig. 3. In either case, TorsinA is envisioned to wrap around a lipid tube bulging from the liposome surface. **b**, **c** Torsin's N-terminal hydrophobic portion is depicted in two alternative states on the reconstructed TorsinA filaments. The first structured residue on torsin's N terminus (TorsinA R58) is marked as a gray circle, whereas the Cys162 is shown as a yellow circle. The N-terminal hydrophobic segment of TorsinA is conceived to be freely moving either toward inside (alternative 1) or outside (alternative 2) of the filaments. The yellow circle on the hydrophobic segment marks Cys49/50, which come in close contact with Cys162 in the inward facing model

membrane-free filaments (Fig. 3g). Assuming a monolayered protein coat, we can estimate a membrane structure with a diameter of ~8.5 nm size (Fig. 6a), since the protein coat thickness itself would be close to the one observed in the membrane-free environment.

As detailed above, our data suggest that TorsinA can assemble into helical filaments of various diameter, thus presumably with different periodicity. Our construct lacked the N-terminal 30 residues of TorsinA (aa 21–50), which remain after signal peptide cleavage and translocation into the ER. This N-terminal peptide is hydrophobic and is suggested to non-specifically interact with membranes mediating torsin's proper ER localization and retention[28]. Its location within the filaments is interesting. Sterically, it could be located on the outside or the inside of the helical tube (Fig. 6b, c). Possibly, the location may be dictated by the redox state of the conserved cysteines of Torsin. Specifically, Cys162 is spatially close to Cys49/50. One can entertain the possibility that a disulfide bridge between Cys49/50 and Cys162 may direct the hydrophobic N terminus to be in the inside of the helical tube. A reduction of this potential disulfide bridge may, in

contrast, direct the N terminus to be on the outside of the helical tube instead. In addition to a redox-regulated mechanism, the strict conservation of multiple cysteines in the vicinity of the hydrophobic N terminus may also point to metal coordination and its exploit for regulation. These are interesting subjects to explore in future studies.

How could the helical filament fit into currently discussed biological functions of TorsinA? Schlieker et al. laid out possible models for TorsinA function in their recent articles[10,29]. Central to this function are the omega-shaped blebs of the INM that are observed in Torsin knockouts[35]. These blebs remain connected with the INM with a nuclear pore-like structure at the neck. TorsinA may be necessary to resolve these necks or else it could be involved in the fusion of the INM and the outer nuclear membrane (ONM). Both scenarios would entail major membrane remodeling steps. While our studies support the notion that TorsinA can indeed remodel membranes, the fact that all of our inner channel mutants still rescue the blebbing phenotype of the triple TorsinA/B/3A KO argues against a direct role of TorsinA filaments in this process (Fig. 4). Another potential caveat is that our filaments were generated with an N-terminally truncated TorsinA construct, and we know that the missing N terminus is important for the rescue of the blebbing phenotype (Supplementary Fig. 11). However, an N-terminally truncated form of TorsinA has been described previously as a result of DTT-induced ER stress[47]. In those experiments, TorsinA gets proteolytically cleaved between residues Cys49 and Cys50, generating a protein just one residue longer than our construct. While the function of this processed form of TorsinA is still unclear, it is tempting to speculate that it may indeed operate in a filamentous form, similar to the assemblies we observed in vitro. Also, several Torsin homologs do not harbor N-terminal, hydrophobic extensions, i.e., Torsin2A and Torsin3A in humans. Therefore, it is also possible that our filament structure points to a specific function of these Torsin homologs. To explore this possibility, Torsin2A/3A filament formation should be tested in the future.

The filament structure poses an interesting problem regarding TorsinA activation. To trigger ATP hydrolysis, TorsinA needs to be activated by LAP1 or LULL1, however, they cannot be integrated into the filament[18,29]. This generates two viable options. In the first scenario, the activator binds to the nucleotide-exposing end of the filament, and hydrolyzes ATP. This could trigger the dissociation of the terminal Torsin molecule, however, it would require an allosteric mechanism. In other words, the penultimate Torsin–Torsin interface would need to "sense" the ATP-to-ADP conversion at the LAP/LULL binding site of the terminal Torsin. Sequentially, the filament could be dissolved this way. Alternatively, LAP1 or LULL1 could randomly insert into the filament, and break it stochastically at various places. This second scenario depends on the filament being dynamic, which can be assumed based on its in vitro properties. The association of LAP1 and LULL1 with TorsinA is stronger than the homotypic TorsinA interaction, therefore the local concentration of the binding partners is an important determinant of the relevant oligomeric state at a given time.

Taken together, we show that TorsinA can form filaments and that it interacts with membranes, even without the ~30 aa hydrophobic N-terminal region present. Going forward, it will be important to further dissect the ATP hydrolysis cycle and to study the protein in situ at higher resolution. TorsinA appears capable of engaging liposome membranes without ATPase activity. However, in the physiological context, this action could be coupled to ATP hydrolysis, raising the possibility that ER/NE membranes are Torsin substrates. No matter where additional findings will lead us, it is obvious that TorsinA and its homologs are AAA + ATPases that continue to defy conventions.

## Methods

**Protein expression and purification.** Human TorsinA (residues 51–332), N-terminally fused with a human rhinovirus 3C protease cleavable MBP tag, was cloned into a modified ampicillin resistant pETDuet-1 vector (EMD Millipore). Mutations were introduced by site-directed mutagenesis.

The *E. coli* strain LOBSTR(DE3) RIL (Kerafast)[48] was transformed with the MBP-TorsinA construct. Cells were grown at 37 °C in the lysogeny broth (LB) medium supplemented with 100 µg ml$^{-1}$ ampicillin, and 34 µg ml$^{-1}$ chloramphenicol until an optical density (OD$_{600}$) of 0.6–0.8 was reached, shifted to 18 °C for 20 min, and induced overnight at 18 °C with 0.2 mM isopropyl β-D-1-thiogalactopyranoside (IPTG). The bacterial cultures were harvested by centrifugation, resuspended in lysis buffer (50 mM HEPES/NaOH pH 8.0, 400 mM NaCl, 10 mM MgCl$_2$, and 1 mM ATP) and lysed with a high-pressure homogenizer (LM20 Microfluidizer, Microfluidics). The lysate was immediately mixed with 0.1 M phenylmethanesulfonyl fluoride (PMSF) (50 µl per 10 ml lysate) and 250 units of TurboNuclease (Eton Bioscience), and cleared by centrifugation. The soluble fraction was gently mixed with amylose resin (New England Biolabs) for 30 min at 4 °C. After washing with lysis buffer, bound protein was eluted in elution buffer (10 mM HEPES/NaOH pH 8.0, 150 mM NaCl, 10 mM MgCl$_2$, 1 mM ATP, and 10 mM maltose).

TorsinA filaments were observed when the eluted protein was diluted in elution buffer without maltose and negatively stained (see below for details). The filaments grew longer after the MBP fusion tags were cleaved with 3C protease during dialysis against a variety of buffer conditions. For cryo-EM structural analysis of TorsinA, the eluted protein (1–1.5 mg/ml) was mixed with 3C protease and dialyzed overnight at 4 °C against 10 mM HEPES/NaOH pH 8.0, 300 mM NaCl, 10 mM MgCl$_2$, and 0.5 mM ATP.

**Liposome preparation.** Stock lipid solutions (Avanti Polar Lipids) were resuspended in chloroform. Small acidic liposomes used in membrane remodeling experiments were produced by combining 65 mole % egg L-α-phosphatidylcholine (PC) and 35 mole % phosphatidylserine di18:1 (DOPS) at 10 mg/ml in polypropylene tubes. The lipid mixes were speed vacuumed to dryness overnight, and the resulting lipid films were rehydrated in liposome buffer (20 mM HEPES/NaOH pH 8.0, 100 mM KCl, 10 mM MgCl$_2$, and 5% w/v sucrose) for 6 h at room temperature with intermittent vortexing. Next, the lipids were subjected to four cycles of rapid freeze/thaw and extruded 15 times (Avanti Mini-Extruder) through 0.2-µm polycarbonate membranes at 65 °C. Liposomes were aliquoted, snap-frozen, and stored at −80 °C.

**Membrane remodeling reactions.** MBP-TorsinA eluted from the amylose resin was diluted to ~0.3 mg/ml and dialyzed against a lower ionic-strength buffer (20 mM HEPES/NaOH pH 8.0, 100 mM NaCl, 10 mM MgCl$_2$, 0.5 mM ATP, and 5% w/v sucrose) to facilitate lipid binding. Initially, liposome mixes were prepared at NaCl concentrations ranging from 80 to 300 mM, but lipid protrusions were observed most abundantly at 80–100 mM NaCl. In all, 2.5–3 µM of TorsinA was mixed with liposomes in a 50 µl final volume at a molar ratio of protein-to-lipid ranging from 1:10 to 1:20. MBP-tagged TorsinA was incubated together with 3C protease and liposomes for 6 h. Alternatively, if 3C protease was included during the dialysis of Torsin, the pre-cleaved TorsinA was incubated with liposomes for an hour. Lipid protrusions were observed regardless of whether the incubations were performed at room temperature or at 4 °C.

**Negative-stain electron microscopy.** Overall, 5 µl of TorsinA filaments or TorsinA–liposome mixtures were loaded on glow-discharged (EMS 100, Electron Microscopy Sciences) continuous carbon film grids (CF200-Cu, Electron Microscopy Sciences) immediately after diluting them. TorsinA filaments were diluted in the same buffer which they were dialyzed against (typically 20 mM HEPES/NaOH pH 8.0, X mM NaCl, 10 mM MgCl$_2$, 0.5 mM ATP). TorsinA–liposome mixtures were also diluted in dialysis buffer (20 mM HEPES/NaOH pH 8.0, 100 mM NaCl, 10 mM MgCl$_2$, 0.5 mM ATP), except lacking sucrose to prevent interference with negative staining. Samples containing only liposomes were prepared identically to TorsinA–liposome mixtures, and the final sucrose concentration after dilution was about 1% w/v. After 45 s of adsorption on grids, the samples were blotted and the specimen on the grid was immediately stained with 2% w/v uranyl acetate for 30 s. The specimen was blotted, stained once more, re-blotted, and air dried. Electron micrographs were recorded on an FEI Tecnai Spirit BioTwin microscope (FEI) operated at 80 keV and equipped with a tungsten filament and an AMT XR16 CCD detector.

**Electron cryomicroscopy.** For electron cryomicroscopy, 3 µl of ~0.06 mg/ml TorsinA polymers were applied to glow-discharged Quantifoil R1.2/1.3 400 mesh Cu holey carbon grids (Quantifoil, Germany), blotted (8 s) and plunge-frozen in liquid ethane using a Vitrobot Mark IV (Thermo Fisher Scientific). Data collection was carried out at liquid nitrogen temperature on a Talos Arctica microscope (Thermo Fisher Scientific) operated at an accelerating voltage of 200 kV. Micrographs were recorded as movie frames at a nominal magnification of 36,000× with underfocus values between 0.5 and 1.5 µm on a K2 Summit direct electron detection camera (Gatan) operated in counting mode (Table 1). During an 8-

second exposure, 40 movie frames were collected, resulting in a total accumulated dose of 30 electrons per Å². Frames were aligned and summed up using the program *AlignFrames* (IMOD).

**Helical reconstruction.** The defocus values and the astigmatism of the micrographs were determined by CTFFIND3[49]. A total of 602 good micrographs were selected based on the CTF estimation and defocus < 3 μm for subsequent image processing. CTF was corrected by multiplying the images with the theoretical CTF, which correct the phases and improves the signal-to-noise ratio. The *e2helixboxer* routine within EMAN2[50] was used for boxing the long filaments from the images. A total of 75,909 384 px-long overlapping segments (with a shift of 8 px between adjacent segments) were extracted from the long filaments for further reconstruction in SPIDER[51]. Using a featureless cylinder as an initial reference, 69,670 segments were used in IHRSR cycles until the helical parameters (a rotation of 42.5° and an axial rise of 5.5 Å per subunit) converged. The resolution of the final reconstruction was determined by the Fourier shell correlation (FSC) between two independent half maps, generated from two non-overlapping data sets, which was 4.4 Å at FSC = 0.143.

**Model building and refinement.** We used the TorsinA crystal structure (PDB ID: 5J1S, chain A) as an initial template to dock into the cryo-EM map by rigid body fitting, and then manually edited the model in UCSF Chimera[52] and Coot[53]. We then used the modified model as the starting template to further refine with the RosettaCM de novo model-building tool[54]. The refined monomeric model of TorsinA was then re-built by RosettaCM with helical symmetry and real-space refined by Phenix[55] to improve the stereochemistry, as well as the model map coefficient correlation. The TorsinA model was validated with MolProbity[56], and the coordinates were deposited to the Protein Data Bank with the accession code 6OIF. The corresponding cryo-EM map was deposited in the EMDB with accession code EMD-20076. The refinement statistics are listed in Table 1.

To calculate the electrostatic potential of the wild-type and the mutant TorsinA filaments, PDB format files were converted to PQR format with the PDB2PQR server[57] using the PARSE force field and assigned protonation states at pH 7.0, and then applied to the APBS program[58] implemented in UCSF Chimera. The evolutionary conservation of the TorsinA polymer was illustrated using the ConSurf server[59] supplied with the multiple sequence alignment that we generated previously[18]. Protein interfaces within the TorsinA filaments were analyzed using the PDBePISA server[60]. Structure figures were created using UCSF Chimera[52] and PyMOL (Schrödinger LLC).

**ATPase activity assay.** ATP hydrolysis rates of TorsinA, TorsinA:LAP1, and TorsinA:liposome mixtures were measured by an NADH-coupled assay as described previously[24]. To remove the trace amount of bacterial GroEL contamination, MBP-TorsinA was purified by size exclusion chromatography on a Superdex S200 column (GE Healthcare) equilibrated in running buffer (20 mM HEPES/NaOH pH 8.0, 100 mM NaCl, 10 mM MgCl₂, 0.5 mM ATP, and 5% w/v sucrose). The luminal portion of human LAP1 (residues 356–583) was purified as described previously[24], and then dialyzed against the same running buffer. The assays were carried out at room temperature in running buffer and in the presence of 2 mM ATP. In all, 3.5 μM of TorsinA and 100 μM of liposomes were used in 30 μl of reaction volume, mimicking the protein-to-lipid molar ratio used in membrane remodeling reactions. In all, 3.5 μM of TorsinA filaments and 3.5 μM of LAP1 were mixed in another reaction as a positive control. The 3C protease was included in all mixtures to provide removal of the MBP tags. The negative control reaction contained all of the components of the assay, except TorsinA. Data analysis was performed using GraphPad (La Jolla, CA) Prism.

**Generation of knockout and rescue cell lines.** HeLa cells were cultured in the Dulbecco's modified Eagle medium (DMEM) supplemented with 10% fetal bovine serum (FBS) and penicillin/streptomycin. Using http://crispr.mit.edu, the following guide sequences were designed to generate a triple knockout (KO) of TorA/B/3 A: TorA, 5′-GCGGGTAGATGTAGCCGGTG-3′; TorB, 5′-CCTAGCCATCGGG GCCGCGT-3′; and Tor3A, 5′-GCGCCACGGACCGCGAAGCA-3′. Cas9 and the single-guide RNAs (sgRNAs) designed against TorA and TorB were expressed in pX330-GFP[61] as described previously[62]. Similarly, Cas9 and the sgRNA designed against Tor3A were expressed in pX330-BFP[61]. Deletion of Torsins were carried out sequentially. First, a TorA KO cell line was generated, and then TorB and Tor3A were simultaneously deleted in that cell line. In each case, cells were transfected with the pX330 constructs using FuGENE HD (Promega) according to the manufacturer's instructions, and clonal populations were isolated by single-cell sorting after 2 days using GFP and/or BFP signal. Deletion of Torsins was verified via immunoblotting (see below for details).

To confirm deletion of Tor3A, we also performed deep sequencing on the genetic locus enclosing the Tor3A target site. Genomic DNA of cell pellets were extracted in lysis buffer (100 mM Tris/HCl pH 8.0, 5 mM EDTA, 200 mM NaCl, 0.2% SDS, supplemented with 0.2 mg/ml Proteinase K from New England Biolabs) at 55 °C overnight. Following precipitation in isopropanol, the DNA pellet was washed with 75% ethanol and resuspended in TE buffer. A ~240 -bp region centered around the CRISPR target site was PCR amplified using the primers

5′-GGGCTTAAGGGAGCC TGGCTAGGCCGG-3′ and 5′-GTCCAGTACCGCTT GGAGAGGGCACCCG-3′. Purified PCR product was submitted to the CCIB DNA Core facility (Massachusetts General Hospital/Harvard) for deep sequencing. A single distinct mutant allele was identified containing a frameshift mutation at the 5′ exon of Torsin3A, and the wild-type allele was absent in the PCR product mixture.

Rescue cell lines stably expressing the TorsinA (aa 1–332)—3xHA constructs were generated by retroviral infection. pBABE-puro plasmids containing the TorsinA-3xHA variants were co-transfected into 293-GP cells along with the VSV-G pseudotyping plasmid for the production of amphotropic retrovirus[63]. The resulting retrovirus was mixed with 20 μg/ml hexadimethrine bromide (Sigma-Aldrich) and incubated with the TorA/B/3 A KO cell line for 2 days. Afterward, cells were split for selection in 0.5 μg/ml puromycin. After 2 weeks of selection, clonal populations were isolated by single-cell sorting, and TorsinA-3xHA expression levels of individual clones were assessed by immunoblotting.

**Immunoblotting.** For immunoblotting, cell pellets were incubated on ice for 30 min in lysis buffer (50 mM Tris/HCl pH 7.5, 150 mM NaCl, 10 mM MgCl₂, 1 mM ATP, 1% IGEPAL CA-630, 0.1% sodium deoxycholate, supplemented with complete (Roche) EDTA-free protease inhibitor tablets and PhosSTOP (Roche) phosphatase inhibitor tablets). The cell lysates were cleared by centrifugation at 15,871×g for 15 min on a benchtop centrifuge at 4 °C. The total protein concentration in lysates was measured with a BCA protein assay kit (Pierce). Following size separation on an SDS-PAGE gel, samples were semi-dry transferred to the nitrocellulose and blocked in a buffer containing 5% w/v skim milk in TBST for 1 h at RT. Primary and secondary antibodies were diluted in a buffer containing 3% w/v BSA in TBST. SuperSignal West Pico PLUS (Thermo Fisher Scientific) was used as the ECL substrate. The antibodies used for immunoblotting were the following: mouse monoclonal D-M2A8 against TorA (a gift from Cristopher Bragg, Massachusetts General Hospital/Harvard) in 1:100; rabbit anti-TorB (a gift from Rose Goodchild, VIB-KU Leuven Center for Brain & Disease Research, Leuven, Belgium) at 1:100; rabbit polyclonal anti-Tor3A (ARP33117_P050, Aviva Systems Biology) at 1 mg/ml; mouse monoclonal anti-vinculin (ab130007, Abcam) at 1:10,000; goat anti-mouse IgG-HRP (sc-2055, Santa Cruz Biotechnology) at 1:5000; and goat anti-rabbit IgG-HRP (7074P2, Cell Signaling Technology) at 1:2000.

To assess the presence of GroEL contamination in purified TorsinA variants, immunoblotting was performed similarly, and a rabbit anti-GroEL (G6532, Sigma-Aldrich) was used at 1:20,000.

**Immunofluorescence microscopy.** Cells were fixed with 4% formaldehyde in phosphate-buffered saline (PBS) buffer for 10 min, and then permeabilized with 0.2% Triton X-100 for 5 min. Blocking and all antibody dilutions were performed in AbDil solution (50 mM Tris/HCl pH 7.5, 150 mM NaCl, 0.1% Triton X-100, 3% bovine serum albumin, and 0.1% NaN3). PBS was used for washes. The primary antibodies used for immunofluorescence were the following: rabbit anti-K48 Ubiquitin (05–1307, Millipore) at 1:500; mouse anti-Lamin A (ab8980, Abcam) at 1:1000; mouse anti-HA (H9658, Sigma-Aldrich) at 1:1250. Anti-mouse Cy3 (715–165–150, Jackson ImmunoResearch) and anti-rabbit Cy2 (711-225-152, Jackson ImmunoResearch)-conjugated secondary antibodies were used at 1:300 dilution. DNA was visualized by incubating cells for 10 min in 1 μg/ml Hoechst-33342 (Sigma-Aldrich) in PBSTx solution (PBS supplemented with 0.1% Triton X-100). Coverslips were mounted using 0.5% p-phenylenediamine and 20 mM Tris/HCl pH 8.8, in 90% glycerol.

Images were acquired on a DeltaVision Core deconvolution microscope (Applied Precision/GE Healthsciences) equipped with a CoolSnap HQ2 CCD camera (Photometrics). For representative fluorescence images a 100 × 1.40 NA Olympus U-PlanApo objective, and for quantitative analysis of the K48-Ubiquitin foci a 40 × 1.35 NA Olympus U-PlanApo objective were used. Five Z-sections were acquired with 0.2 -μm spacing, and images were deconvolved using the DeltaVision software.

**Ultrastructural analysis of cell lines.** Cells were fixed in 0.1 M sodium cacodylate (pH 7.4) buffer containing 2.5% glutaraldehyde, 3% paraformaldehyde and 5% sucrose, pelleted, and post fixed in 1% OsO₄ in veronal-acetate buffer. Next, they were stained en bloc overnight with 0.5% uranyl acetate in veronal-acetate buffer (pH6.0), dehydrated, and embedded in Embed-812 resin. Sections cut on a Leica EM UC7 ultra microtome with a Diatome diamond knife at a thickness setting of 50 nm were stained with 2% uranyl acetate, and lead citrate. The sections were then examined using a FEI Tecnai spirit BioTwin microscope (FEI) at 80 keV and photographed with an AMT XR16 CCD camera. About 30 EM cross-sections per cell line were imaged for morphometric analysis of the nuclear envelope. The NE membrane enclosed in each image was measured with Fiji[64], and the quantitative analysis of the number of nuclear blebs was performed using GraphPad (La Jolla, CA) Prism.

**Reporting summary.** Further information on research design is available in the Nature Research Reporting Summary linked to this article.

## Data availability

All data and the biologically unique materials generated in this study are available from the corresponding author upon request. A reporting summary for this article is available as a Supplementary Information file. The atomic coordinates for the TorsinA structure have been deposited in the Protein Data Bank (PDB) under the accession code 6OIF, and the corresponding EM density map has been deposited to the Electron Microscopy Data Bank (EMDB) under the accession code EMD-20076. The source data underlying Figs 3, 4, and Supplementary Figs 5, 7, 8, 10 are provided as a Source Data file.

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

## Acknowledgements

We thank Xu Chen and KangKang Song for help with cryo-EM data collection at the UMass Medical Center, Worcester cryo-EM facility. We thank Ed Brignole for advice on cryo-EM data collection and processing in the initial stage of the project. This work was supported by NIH AR065484 (to T.U.S.) and R35GM122510 (to E.H.E.). This work was further supported by the U.S. Army Medical Research Acquisition Activity (AMRAA), through the Peer Reviewed Medical Research Program (PRMRP) under Award No W81XWH1810515 (to T.U.S.).

## Author contributions

F.E.D. and T.U.S. designed the study; F.E.D. performed the biochemical, structural, and cell biological experiments; W.Z. and E.H.E. performed image analysis and helical reconstruction; A.J.M. and V.D. helped with membrane tubulation assays; N.M. and I.M.C. helped with generation of knockout and rescue cell lines and fluorescence microscopy; N.W. performed sectioning and ultrastructural analysis of cell lines; F.E.D. and T.U.S. analyzed and interpreted the data; all authors discussed the results and contributed to the final paper.

## Additional information

**Competing interests:** The authors declare no competing interests.

