## [Peer Review File · Nature Communications]

Reviewers' comments:

Reviewer #1 (Remarks to the Author):

This manuscript describes the cryo-EM structure of a filamentous form of the AAA+ ATPase TorsinA. The structure shows the nature of protomer-protomer interactions, and describes a conformational change within the protomer that gives rise to a different helical symmetry in the filament than in existing crystal structures. The paper also describes tubulation of lipid membranes by TorsinA, and suggests a model for electrostatic interactions between phospholipids and alternating rings of positively charged residues lining the internal surface of the TorsinA filament.

The cryo-EM work is well done, and clearly presented. Although the final resolution of the TorsinA structure is relatively modest (4.4 Å), the overall conclusions as regards filament geometry and protomer conformation are well supported by the cryo-EM density - that is to say, the authors draw no conclusions that go beyond what is supported at this resolution. Overall, the work is interesting and in my view merits publication after addressing a few minor issues:

1) ATPase activity assay (Supplementary Figure 5): While the three conditions clearly exhibit virtually no activity, a positive control should be included to provide the reader confidence that the assay is functioning properly. Are there conditions known to promote ATPase activity in TorsinA that could be used? Alternatively, a different ATPase of known activity should be used.

2) polymerization blocking mutations (Supplementary Figs. 4 &5): Mutations at the lateral and longitudinal interfaces in the helices clearly disrupt polymer formation. However, the mutations at the lateral interfaces would be anticipated to completely block assembly, but it appears from the negative stain images that much (most?) of the protein is in small oligomers (possibly hexamers?). This seems inconsistent with the idea that the mutations are blocking assembly interactions - can the authors explain this observation?

Reviewer #2 (Remarks to the Author):

In their manuscript titled "The AAA+ ATPase TorsinA polymerizes into hollow tubes with a helical periodicity of 8.5 subunits per turn," Demircioglu et al. report that a truncated TorsinA construct self-assembles into filamentous structures. Using cryo-electron microscopy, the group captures the first higher resolution images of TorsinA homo-oligomerization. This is a considerable advancement that broadens our understanding of AAA+ ATPases significantly. The authors also put forth additional data suggesting TorsinA participates in membrane extrusion, a function that is likely to be distinct from TorsinA filamentation. This provocative idea is, while highly original, in the early stages of development. Nonetheless, it will motivate the field to further explore the relationship between TorsinA and membranes. The insights presented in this manuscript are of high interest to the field and advance our current understanding of this enigmatic AAA+ ATPase. We recommend this manuscript for publication in Nature Communications if the authors can address the following concerns:

Major concerns:

1. As truncated constructs were used for filament formation and membrane extrusion experiments, it is somewhat difficult to directly compare the in vitro behavior of the truncated constructs with the in vivo behavior of full-length proteins (note that the N-terminal hydrophobic domain of TorsinA is critical for membrane anchoring of TorsinA and for proper localization; PMID: 21785409). Have the authors tested full-length, membrane-integrated TorsinA for filament formation? Alternatively, would truncated

delta51 constructs rescue the KO phenotype (see also minor concern # 4)?

2. For the main text, it would be useful for the reader to provide a schematic model illustrating the arrangement of the TorsinA filament interacting with a membrane. This model should clarify how the authors believe the TorsinA-coated membranes are organized, and how TorsinA is oriented in the membrane protrusions. This would also help the reader to better relate diameter measurements with the membrane-bound and membrane-free structures.

3. The authors utilize a 3C protease treatment to generate free TorsinA protein. However, the mixture of unprocessed MBP-TorsinA and free MBP that was apparently used in some experiments increases the overall heterogeneity of the sample, which complicates the interpretations of their *in vitro* analyses. Specifically, the authors cannot definitively conclude that membrane protrusions result from uniform TorsinA or MBP-TorsinA assemblies, or a mixture of the two species. Since MBP is significantly larger than TorA, this complication factor can affect size/diameter measurements considerably. This should be stated in the manuscript, or alternatively resolved by additional experiments by using more homogenous preparations.

4. While the authors acknowledge that the filaments and membrane extrusion are "at least partially independent of each other", there are no data presented suggesting the two would be related. In fact, the mutational analyses argue against a direct functional significance of these phenomena in the context of NE blebbing/NPC biogenesis. It is not clear to this reviewer why the authors still attempt to connect the two in the discussion. Doesn't this study instead suggest a new activity/function (representing a highly original, novel spin on the Torsin field)? Perhaps this part of the discussion could be rephrased to tame down some of the correlations while putting more emphasis on the distinct novelty of the presented findings?

5. The discussion would benefit from a schematic model to better convey the topological constraints and membrane association of all players, including LAP1/LULL1. This figure, perhaps also highlighting the location of Cys residues, would make the discussion more accessible to the outsiders of the Torsin field.

Minor concerns:

1. The authors suggest that the reason the TorsinA E171Q mutant does not form filaments is because ATP is bound in a non-canonical or transition state. In the opinion of this reviewer, a stable transition state (by definition a short-lived, high energy state) capable of supporting filament formation would be quite unusual. It would probably be better to avoid the term "transition state" in this context.

2. Throughout the text, the authors should consistently specify whether values reflect the inner or outer diameter of various structures.

3. The authors should expand Table 2 to include the E171Q mutant and full-length TorsinA (see major concern #1).

4. The TorsinA truncation used in the present manuscript is nearly identical to a biologically relevant delta49 truncation known to occur in response to ER stress (PMID: 26953341). Have the authors considered the possibility that the filament may form in the ER upon TorsinA processing to fulfil a filamentation-related function that is distinct from NE blebbing/NPC biogenesis? This could potentially explain the disconnect between the *in vitro* analysis of mutants and the KO complementation data. Perhaps this possibility could be explored in the discussion?

5. Labels and amino acid numbers should be added to Figure S6; it is otherwise difficult to pinpoint

the mutated residues in the electrostatic maps. It would also be useful for future reference to add a sequence alignment and mark the corresponding residues modified to generate inner cavity mutants (as part of the same figure).

6. The authors state that the wild type and mutant TorsinA constructs rescue nuclear blebbing, however they do not state how this is assessed. Are the data as “black-and-white” as suggested by the provided EM images? Some basic form of quantification/statistical analysis would be desirable.

Reviewer #3 (Remarks to the Author):

The TorsinA AAA+ ATPase, which is implicated in the genetic disease primary dystonia and is thought to perform a membrane-associated function, differs from other family members in its dependence upon an activating partner protein to provide the trigger arginine residue that completes the active site. The substrate(s) and mechanism of Torsin function are not known. The current study reports a helical reconstruction of TorsinA that has 8.5 subunits per turn and a large inner cavity of 4nm diameter. This is a distinctly different architecture from other AAA+ ATPases, which are generally active as hexamers.

Mutagenesis and liposome-binding and remodeling assays are also reported. These show interesting properties, but do not support a direct role for the determined helical assembly or show a correlation with a nuclear blebbing assay.

Enthusiasm is tempered by the lack of a clear connection between the helical conformation observed and a biological activity or mechanism, especially because some other AAA+ ATPases have been shown to adopt “non-functional” assemblies in the absence of bound substrate. Nevertheless, this manuscript reports intriguing observations that should be on the radar of investigators thinking about Torsins and other AAA+ ATPases, and the helical structure will prompt further experiments to test models of Torsin function.

There are no technical concerns. The manuscript is clearly written and illustrated. I would like to see it published.

Chris Hill

Reviewer #4 (Remarks to the Author):

The manuscript “The AAA+ ATPase TorsinA polymerizes into hollow tubes with a helical periodicity of 8.5 subunits per turn” by Demircioglu and co-authors reports on the biochemical and structural studies of TorsinA, a protein belonging to the AAA+ ATPases. However, it has a very specific feature as it needs an external activator to function.

The authors have used biochemistry and cryo-EM in their research. They found that TorsinA forms helical filaments in vitro, the structure of which was analysed by helical reconstruction. The reconstruction revealed that the protein forms helices with 8.5 subunits per turn, and a central channel of ~4 nm diameter. Experiments where acidic liposomes were complexed with TorsinA under various conditions have shown that at low ionic strength the liposomes have numerous protrusions of larger diameter (~ 20 nm) than the membrane-free TorsinA filaments (~15.5 nm). At high ionic

strength no protrusions were observed. The authors suggest that in vivo, this membrane/TorsinA interaction could be coupled to ATP hydrolysis. The reported results show that this atypical AAA+ ATPase forms helical assemblies that may interact with cell membranes.

The EM structure provides information on the helical parameters of the filaments and allowed the X-ray atomic coordinates of TorsinA (5J1S) to be fitted into the map. The fitting indicates that conformation of TorsinA in the filament is incompatible with an activator LULL1 interaction. This is interesting research that alludes to a curious hypothesis that will need further extensive study.

This MS has raised the following questions:

- It is unclear (introduction, middle part) what the authors intend to say: “mechanical interaction with the substrate...”. Typically, any interaction between bio complexes is not mechanical but is based on chemical or electrostatic interactions.
- It would be beneficial to show more explicitly why homology between the Clp protein and TorsinA is misleading. The differences have to be present in their sequences and therefore reflected in their structures. This would explain the differences in function. Please clarify that point.
- The authors write that LAP1 and LULL1 are required to activate TorsinA. Both these activators appear to be transmembrane proteins. A reference is needed here. This fact indicates that maybe TorsinA should be bound to the membrane or one of these activators, to be in its functional state. However, throughout the MS it remains unclear what is the major function of TorsinA.
- It seems that oligomerisation of TorsinA takes place exclusively in the conditions where no other proteins are present (see the end of the paragraph on the second page of the Introduction) but this is not similar to other AAA+ ATPases.
- It would be useful to show the location of G251 in Suppl. Fig 3a (left panel). It is difficult to understand from the right panel why mutation of this residue will prevent oligomerisation. What sort of interaction takes place here? It is unclear from the MS.
- How was the polarity of the possible TorsinA tubes attached to the membrane defined? How were the hypotheses of interactions between tubes and membranes verified? It seems both ends of the filament have rings with opposite charges so they can interact either way with the liposomes.
- One completely agrees with the authors that the resolution of the structure obtained is rather low: about 7-8 Å. They have to use a resolution at the 0.5 threshold of FSC (that has a slightly funny shape) which will more realistically reflect the quality of the structure. A reader can easily see some elongated densities corresponding to alpha helices, but not all of them are well defined. None of them show the grooves in alpha helices that should be seen at a resolution of ~ 5 Å. Beta layers within the EM structure are not resolved: at a resolution of 4.5 Å the strands of beta layers should be separated. Another confusing issue is related to the diffraction pattern of the TorsinA helix (Suppl. Fig 2b). If the pitch of the helix is ~ 47 Å, then the diffraction pattern only shows reflections up to ~ 16 Å. Therefore, it seems that the authors did not have data that would give a resolution of 4.5 Å. This low resolution prevents the authors from making reasonable conclusions on interactions between subunits within the helix. It would be good to have the helices numbered as it would make it easier both for the description and the reader.
- How have the authors verified that TorsinA forms the outer layer of the liposome protrusion? It would be essential for the authors to perform experiments with FABs for TorsinA. It could be

suggested that the liposome membrane covers the TorsinA tube. Otherwise, how can the authors explain the roundish ends of protrusions?

- It would be helpful if the authors explain what is to be seen in the Figure 4c,d,e,f. The figure legend does not provide any explanation.
- It would be important to check what is the role of aa 21-50 that were not in the construct. Are they crucial for the specific activity of this protein?

Minor comments:

- Why do the authors use the term "cavity" for the central channel in the TorsinA helix? Possibly "channel" would be more suitable, since "cavity", according to the dictionary, means a hollow or some empty space within the solid object, and typically has only one entrance, that also works as an exit. Possibly tunnel or channel would be better used to describe the empty internal part of the helix along the central axis.
- End of the introduction. This sentence is confusing: "the hollow core the these filaments may bind directly to membranes...". How could something which is empty, bind anything?
- It would be better to use the term "segment" instead of "particle" throughout the MS and methods when the authors are writing about image processing and selecting fragments of filaments to process.
- "wide open channel" – can a channel be closed? In theory, yes, if it is blocked by something, but that situation was not discussed in this MS.

We thank the reviewers for their fair and constructive criticism of our manuscript. In this revised version we have tried to address the comments as best as we could. Here are our point-by-point responses:

Reviewer #1:

- 1) *ATPase activity assay (Supplementary Figure 5): While the three conditions clearly exhibit virtually no activity, a positive control should be included to provide the reader confidence that the assay is functioning properly. Are there conditions known to promote ATPase activity in TorsinA that could be used? Alternatively, a different ATPase of known activity should be used.*

Response:

We have previously used the exact same assay in PMID: 25149450. In that paper we showed that LAP1/LULL1 activate TorsinA. This should suffice to give the reader confidence about the assay. We adjusted the figure legend to explicitly refer to our previous work, in addition to the reference to the assay per se.

- 2) *Polymerization blocking mutations (Supplementary Figs. 4 &5): Mutations at the lateral and longitudinal interfaces in the helices clearly disrupt polymer formation. However, the mutations at the lateral interfaces would be anticipated to completely block assembly, but it appears from the negative stain images that much (most?) of the protein is in small oligomers (possibly hexamers?). This seems inconsistent with the idea that the mutations are blocking assembly interactions - can the authors explain this observation?*

Response:

The reviewer is absolutely correct in pointing out the small oligomers that are clearly visible in the negative stain analysis. We have explicitly stated in the figure legend to Supplementary Figure 4 (new Supplementary Figure 5) that these small oligomers in all likelihood represent GroEL contamination from the preparation. In fact, that is why we incorporated the magnified image in panel b that shows that these particles are likely heptameric, further supporting the notion that these particles represent GroEL. Therefore, we believe that we have already addressed this fully in the original manuscript.

Reviewer #2:

1. *As truncated constructs were used for filament formation and membrane extrusion experiments, it is somewhat difficult to directly compare the in vitro behavior of the truncated constructs with the in vivo behavior of full-length proteins (note that the N-terminal hydrophobic domain of TorsinA is critical for membrane anchoring of TorsinA and for proper localization; PMID: 21785409). Have the authors tested full-length, membrane-integrated TorsinA for filament formation? Alternatively, would truncated delta51 constructs rescue the KO phenotype (see also minor concern # 4)?*

Response:

This is a valid point and we have tried to address this further. We have now performed the rescue experiment in the TorsinA/B/3A triple-KO using TorsinA delta 26-43, a construct that was used by the Hanson lab in PMID:21785409. We were not able to rescue the blebbing phenotype, as judged by immunofluorescence using anti-K48 linked Ubiquitin antibody as a marker for nuclear blebs (assay performed as described in PMID: 27798237). The data has been added to the supplement (new Supplementary Figure 11). While this assay shows that the hydrophobic element is likely necessary for the rescue of blebbing, it does not address whether or not the filaments occur *in vivo*. However, it suggests that the filament form is not an element involved in the blebbing phenotype. We have adjusted the discussion accordingly.

We have also tested a full-length MBP-TorsinA fusion protein for filament formation *in vitro*. Unfortunately, in our hands this protein is rather ill-behaved, i.e. it elutes in the void of a Superdex 200 gel filtration column, and it shows aggregates in negative-stain EM analysis. Therefore, we cannot definitively conclude whether or not the full-length TorsinA is able to form filaments. Based on the available structural data we strongly believe that the full-length TorsinA should be able to also form filaments under appropriate conditions. There is no reasonable steric or electrostatic argument that would argue against the full-length protein being able to form filaments as well.

- 2. For the main text, it would be useful for the reader to provide a schematic model illustrating the arrangement of the TorsinA filament interacting with a membrane. This model should clarify how the authors believe the TorsinA-coated membranes are organized, and how TorsinA is oriented in the membrane protrusions. This would also help the reader to better relate diameter measurements with the membrane-bound and membrane-free structures.*

Response:

We have included the new Figure 6 that shows how we envision Torsin to interact with membrane. It is indeed quite useful to do so and we thank the reviewer for making this point.

- 3. The authors utilize a 3C protease treatment to generate free TorsinA protein. However, the mixture of unprocessed MBP-TorsinA and free MBP that was apparently used in some experiments increases the overall heterogeneity of the sample, which complicates the interpretations of their in vitro analyses. Specifically, the authors cannot definitively conclude that membrane protrusions result from uniform TorsinA or MBP-TorsinA assemblies, or a mixture of the two species. Since MBP is significantly larger than TorA, this complication factor can affect size/diameter measurements considerably. This should be stated in the manuscript, or alternatively resolved by additional experiments by using more homogenous preparations.*

Response:

We have repeated the membrane tubulation assay with MBP-TorsinA not treated with 3C protease. The result is that the diameter of the decorated membrane protrusions is comparable to figure 3, consistent with the notion that the MBP-tag does not interfere with lipid interaction (new Supplementary Figure 8). In contrast to the observation in Figure 3, the protrusions

decorated with the MBP-TorsinA have similar diameters to the membrane-free MBP-TorsinA filaments. This indicates that MBP-TorsinA is less likely to arrange into wider filaments on the membrane, presumably due to steric reasons. Since our experiments shown in Figure 3 were performed with a heterogeneous mixture of MBP-TorsinA and untagged TorsinA, the mean diameter of the protrusions that we observed could have been underestimated. In sum, we can exclude that our originally observed protrusions were an MBP-mediated artifact, but we cannot definitively say whether the originally observed protrusions are decorated with purely MBP-free TorsinA. In all likelihood, it is a mixture of cleaved and uncleaved protein.

4. *While the authors acknowledge that the filaments and membrane extrusion are “at least partially independent of each other”, there are no data presented suggesting the two would be related. In fact, the mutational analyses argue against a direct functional significance of these phenomena in the context of NE blebbing/NPC biogenesis. It is not clear to this reviewer why the authors still attempt to connect the two in the discussion. Doesn't this study instead suggest a new activity/function (representing a highly original, novel spin on the Torsin field)? Perhaps this part of the discussion could be rephrased to tame down some of the correlations while putting more emphasis on the distinct novelty of the presented findings?*

Response:

This is indeed a very valid point and we are glad the reviewer articulates it. Our data, including the new blebbing rescue assay with truncated TorsinA is much more consistent with a different, novel function of TorsinA. We have reworded the discussion accordingly.

5. *The discussion would benefit from a schematic model to better convey the topological constraints and membrane association of all players, including LAP1/LULL1. This figure, perhaps also highlighting the location of Cys residues, would make the discussion more accessible to the outsiders of the Torsin field.*

Response:

We have included a model (Figure 6a) depicting a sterically reasonable interaction of the inner channel with lipids forming a lipid nanotube.

minor concerns:

1. *The authors suggest that the reason the TorsinA E171Q mutant does not form filaments is because ATP is bound in a non-canonical or transition state. In the opinion of this reviewer, a stable transition state (by definition a short-lived, high energy state) capable of supporting filament formation would be quite unusual. It would probably be better to avoid the term “transition state” in this context.*

Response:

We now consistently refer to the ATP conformational state in the observed filaments as potentially non-canonical.

2. Throughout the text, the authors should consistently specify whether values reflect the inner or outer diameter of various structures.

Response:

We have addressed this issue in the revised manuscript.

3. The authors should expand Table 2 to include the E171Q mutant and full-length TorsinA (see major concern #1).

Response:

Data included as suggested.

4. The TorsinA truncation used in the present manuscript is nearly identical to a biologically relevant delta49 truncation known to occur in response to ER stress (PMID: 26953341). Have the authors considered the possibility that the filament may form in the ER upon TorsinA processing to fulfil a filamentation-related function that is distinct from NE blebbing/NPC biogenesis? This could potentially explain the disconnect between the in vitro analysis of mutants and the KO complementation data. Perhaps this possibility could be explored in the discussion?

Response:

A valid point that we addressed in the discussion of the revised manuscript.

5. Labels and amino acid numbers should be added to Figure S6; it is otherwise difficult to pinpoint the mutated residues in the electrostatic maps. It would also be useful for future reference to add a sequence alignment and mark the corresponding residues modified to generate inner cavity mutants (as part of the same figure).

Response:

We have reorganized Supplementary Figure 6 (new Supplementary Figure 9) to include a second spacefilling model for each mutant, with the mutation highlighted. This should help orient the reader.

6. The authors state that the wild type and mutant TorsinA constructs rescue nuclear blebbing, however they do not state how this is assessed. Are the data as “black-and-white” as suggested by the provided EM images? Some basic form of quantification/statistical analysis would be desirable.

Response:

We have now quantified our observations and present the data in panel g of Figure 4.

Reviewer #4:

1. *It is unclear (introduction, middle part) what the authors intend to say: “mechanical interaction with the substrate...”. Typically, any interaction between bio complexes is not mechanical but is based on chemical or electrostatic interactions.*

Response:

We have clarified this issue in the revised manuscript. We meant to say that AAA+ ATPases typically engage with substrate to perform mechanical work (i.e. unfold a protein, remodel RNA).

2. *It would be beneficial to show more explicitly why homology between the Clp protein and TorsinA is misleading. The differences have to be present in their sequences and therefore reflected in their structures. This would explain the differences in function. Please clarify that point.*

Response:

The key differences between TorsinA and Clp proteins have been pointed out in previous publications, notably in our 1.4Å TorsinA-LULL1 complex structure (PMID:27490483). Rather than restating the findings we refer the reader to the previous work.

3. *The authors write that LAP1 and LULL1 are required to activate TorsinA. Both these activators appear to be transmembrane proteins. A reference is needed here. This fact indicates that maybe TorsinA should be bound to the membrane or one of these activators, to be in its functional state. However, throughout the MS it remains unclear what is the major function of TorsinA.*

Response:

The function of TorsinA remains largely enigmatic, therefore we simply cannot refer to a validated functional state, other than by reference to canonical AAA+ ATPases, with all the limitations of that approach. Yes, LAP1 and LULL1 are transmembrane proteins, properly addressed as such in the cited references. We now reiterated this in the text as well.

4. *It seems that oligomerisation of TorsinA takes place exclusively in the conditions where no other proteins are present (see the end of the paragraph on the second page of the Introduction) but this is not similar to other AAA+ ATPases.*

Response:

We believe that we have addressed the differences to canonical AAA+ ATPases sufficiently enough. We have reworded statements about the filament formation in places where this may have not been entirely clear.

5. *It would be useful to show the location of G251 in Suppl. Fig 3a (left panel). It is difficult to understand from the right panel why mutation of this residue will prevent oligomerisation. What sort of interaction takes place here? It is unclear from the MS.*

Response:

We believe that the open book representation chosen in Supplementary Figure 3a/b (new Supplementary Figure 4a/b) is the best way to illustrate residues within the binding surface. Therefore, we refrained from labeling the left panels more extensively, which would otherwise obscure the view. Also, we explained in the main text why G251 is an important site within the TorsinA-TorsinA interface and why its mutation disrupts the interaction (independently tested by the Schlieker lab as well, as we referenced).

- 6. How was the polarity of the possible TorsinA tubes attached to the membrane defined? How were the hypotheses of interactions between tubes and membranes verified? It seems both ends of the filament have rings with opposite charges so they can interact either way with the liposomes.*

Response:

We have not determined the polarity of the TorsinA filaments. We believe that this is beyond the scope of this study. The hypothesis that the filaments enclose a lipid tube is based on the unambiguous observation that incubating liposomes with TorsinA results in protein-coated protrusions (Figure 3 c/d/e/f). We have now included a model of how we envision the lipid molecules to line the central channel of the TorsinA filaments (Figure 6a). We believe that this is the most parsimonious explanation of our data. Further than that we have not verified the data. Again, we believe that further verification goes beyond the study of this investigation.

- 7. One completely agrees with the authors that the resolution of the structure obtained is rather low: about 7-8 Å. They have to use a resolution at the 0.5 threshold of FSC (that has a slightly funny shape) which will more realistically reflect the quality of the structure. A reader can easily see some elongated densities corresponding to alpha helices, but not all of them are well defined. None of them show the grooves in alpha helices that should be seen at a resolution of ~ 5 Å. Beta layers within the EM structure are not resolved: at a resolution of 4.5 Å the strands of beta layers should be separated.*

Response:

We disagree that the resolution of the structure is 7-8 Å, and apologize for the inadequate presentation in the original version where the true resolution was not evident. We have modified the paper to include this figure (new Supplementary Figure 3), which shows clearly resolved β -strands (left) and grooves in α -helices (right). The hand of the α -helices is very clear at the resolution we obtained.

- 8. Another confusing issue is related to the diffraction pattern of the TorsinA helix (Suppl. Fig 2b). If the pitch of the helix is ~ 47 Å, then the diffraction pattern only shows reflections up to ~ 16 Å. Therefore, it seems that the authors did not have data that would give a resolution of 4.5 Å. This low resolution prevents the authors from making reasonable conclusions on interactions between subunits within the helix. It would be good to have the helices numbered as it would make it easier both for the description and the reader.*

Response:

There is no simple relation between the furthest reflection seen in the power spectrum (diffraction pattern) and the resolution of the reconstruction. The averaged power spectrum that was shown is an incoherent average generated by adding together the intensities of the power spectra from each of the segments. The reconstruction involves a coherent averaging, where segments are aligned to each other and added so that in Fourier space amplitudes are being added together. This is not like crystallography, where the diffraction pattern establishes the limit in resolution. To give some published examples, the SIRV2 virus reconstruction (DiMaio et al., 2015) was described as $\sim 4 \text{ \AA}$ overall, with some better internal regions at 3.8 \AA . The averaged power spectrum from the images (Supp. Fig. 1 of that paper) had the furthest reflection at $\sim 7.5 \text{ \AA}$. For the SFV1 virus (Liu et al., 2018) the resolution was described as 3.7 \AA , but the furthest reflection seen in the averaged power spectrum (Supp. Fig. 3 in that paper) was at $\sim 5.9 \text{ \AA}$.

DiMaio, F., Yu, X., Rensen, E., Krupovic, M., Prangishvili, D., and Egelman, E.H. (2015). A Virus that Infects a Hyperthermophile Encapsidates A-Form DNA. *Science* 348, 914-917.

Liu, Y., Osinski, T., Wang, F., Krupovic, M., Schouten, S., Kasson, P., Prangishvili, D., and Egelman, E.H. (2018). Structural conservation in a membrane-enveloped filamentous virus infecting a hyperthermophilic acidophile. *Nature Communications* 9, 3360.

9. *How have the authors verified that TorsinA forms the outer layer of the liposome protrusion? It would be essential for the authors to perform experiments with FABs for TorsinA. It could be suggested that the liposome membrane covers the TorsinA tube. Otherwise, how can the authors explain the roundish ends of protrusions?*

Response:

We find it sterically inconceivable that TorsinA would be on the inside of the protrusions. If we start with empty liposomes and add TorsinA to said preparation, TorsinA by definition will be on the outside. Liposomes were not generated in the presence of TorsinA.

10. *It would be helpful if the authors explain what is to be seen in the Figure 4c,d,e,f. The figure legend does not provide any explanation.*

Response:

We have included a Figure legend and refer to this Figure in the text. We believe that we pointed out well enough what the rationale behind doing the experiment was.

11. *It would be important to check what is the role of aa 21-50 that were not in the construct. Are they crucial for the specific activity of this protein?*

Response:

See response to reviewer #2, major concern #1 above.

Minor comments:

- *Why do the authors use the term “cavity” for the central channel in the TorsinA helix? Possibly “channel” would be more suitable, since “cavity”, according to the dictionary, means a hollow or some empty space within the solid object, and typically has only one entrance, that also works as an exit. Possibly tunnel or channel would be better used to describe the empty internal part of the helix along the central axis.*

Response:

We have reworded the text and eliminated the use of the word ‘cavity’ to describe the central channel.

- *End of the introduction. This sentence is confusing: “the hollow core the these filaments may bind directly to membranes...”. How could something which is empty, bind anything?*

Response:

We reworded the sentence.

- *It would be better to use the term “segment” instead of “particle” throughout the MS and methods when the authors are writing about image processing and selecting fragments of filaments to process.*

Response:

We reworded the manuscript accordingly.

- *“wide open channel” – can a channel be closed? In theory, yes, if it is blocked by something, but that situation was not discussed in this MS.*

Response:

We reworded the passage to avoid confusion.

Reviewers' comments:

Reviewer #1 (Remarks to the Author):

The responses to this reviewer's minor concerns in the initial submission have not been addressed, as detailed below.

1) ATPase activity assays: The initial concern was simply that the assay, which showed essentially zero activity under any of the test conditions, lacked a positive control. The authors' response, that experiments published five years ago should serve as a positive control for the current study, is insufficient. Clearly, any biochemical assay that generates only negative results requires a positive control, run in parallel, to provide any assurance that the experiment procedures are working.

2) Polymerization blocking mutations: The initial concern was that point mutants predicted to block any assembly still appeared to be larger oligomers in negative stain images. The author's assertion that the oligomeric species are likely GroEL raises a major concern about the quality of their mutant protein preparations. While some background of the larger species are the primary structures observed in the micrographs in Suppl. Fig. 5. Such a large contamination with GroEL suggests there may be serious problems with the folding or stability of the mutants, which would suggest that they are assembly-defective for reasons unrelated to their role at assembly interfaces. SDS-PAGE gels and gel filtration traces for each of the purified mutants should be shown, or some other indication of the purity of the proteins and their structural state should be presented.

Reviewer #2 (Remarks to the Author):

The authors have constructively addressed our concerns and the revised manuscript has improved considerably over the original version. These modifications also made the manuscript more accessible for a broader audience. We therefore recommend publication in the present form.

Reviewer #3 (Remarks to the Author):

I have no concerns with the revised manuscript, and support publication at this time.

Chris Hill

Reviewer #4 (Remarks to the Author):

The revised manuscript by Demircioglu and co-authors has been improved. Some answers however should be clarified.

Answer to the first reviewer:

The presented image is not convincing: the molecular images do not look like GroEL: side view should have four layers (or even more if it will be a complex with GroES) , the end views should have 7-fold symmetry. The presented views have apparently a 6-fold symmetry, but to be sure, the authors were

supposed to do symmetry analysis of end views. From the presented figure it is rather difficult to assess sizes of the molecules, there are other smaller complexes. SDS or native gel will help to identify which species are existing in the sample. (Supl. Figure 5b). So the issue was not addressed.

What is shown on other panels of Supl. Figure 5? One can hardly see anything: particles are so small. The particle images in other panels have to be analyzed and class averages should be shown with sizes.

Answers to the fourth reviewer:

A2. The answer is not satisfactory; possibly a figure would help.

A6. Yes, it seems to prove the polarity of the TorsinA tubes at binding to membranes will be difficult. These experiments need a bit more time and thinking. However, the claim, that this is an unambiguous fact, that protrusions are covered by the protein, is extremely overrated. That was not proved by any experiments, which are not difficult to do.

A7. Thanks to the authors, they have improved the figures and have shown that a resolution is around 5A.

A8. This diffraction pattern is still very confusing; it is advisable to remove it from the MS: the figure is not informative and misleading. The authors have many other good and interesting figures.

The coherent result of the reconstruction has to show the improved diffraction pattern that should be comparable with the diffraction pattern from the TorsinA tube. These diffractions are shown. For comparison see papers by N. Unwin; the recent one is in IUCrJ, 2017, V 4(4):393-399, <https://doi.org/10.1107/S2052252517005243> .

A9. It is an interesting hypothesis that lipids are located inside of the tube. If the authors are so sure of that the protein is located on outer surface of protrusions, the fab labeling will be rather easy and will remove any ambiguities. The author's answer does not explain why the protrusions have roundish ends and smooth curved transitions between the liposome and the protrusions. It seems that lipids are located on the outer surface, then the model shown in Figure 6 is not correct. It would be sad if the respectful authors will make a mistake. May be lipids sit between protein subunits?

Comments: "Our in vitro experiments performed with liposomes suggest that the inner channel of these filaments may bind directly to lipids, likely relevant to elucidating the enigmatic function of TorsinA."

The correction of the previous version of the sentence should still be improved. The authors have to indicate which parts of the tube wall may bind lipids. The channel does not bind anything by itself.

Reviewers' comments:

Reviewer #1 (Remarks to the Author):

The responses to this reviewer's minor concerns in the initial submission have not been addressed, as detailed below.

1) ATPase activity assays: The initial concern was simply that the assay, which showed essentially zero activity under any of the test conditions, lacked a positive control. The authors' response, that experiments published five years ago should serve as a positive control for the current study, is insufficient. Clearly, any biochemical assay that generates only negative results requires a positive control, run in parallel, to provide any assurance that the experiment procedures are working.

Response:

We have repeated the ATPase activity assay, now including a positive control. The result is as expected, i.e. positive control worked, the other samples showed the same behavior as before. Therefore, our initial conclusions remain valid.

2) Polymerization blocking mutations: The initial concern was that point mutants predicted to block any assembly still appeared to be larger oligomers in negative stain images. The author's assertion that the oligomeric species are likely GroEL raises a major concern about the quality of their mutant protein preparations. While some background of The larger species are the primary structures observed in the micrographs in Suppl. Fig. 5. Such a large contamination with GroEL suggests there may be serious problems with the folding or stability of the mutants, which would suggest that they are assembly-defective for reasons unrelated to their role at assembly interfaces. SDS-PAGE gels and gel filtration traces for each of the purified mutants should be shown, or some other indication of the purity of the proteins and their structural state should be presented.

Response:

We have now included SDS-PAGE analysis for all mutants tested (Suppl. Fig. 5g,h). We have further performed a Western blot to confirm that we do have GroEL contamination in all our samples, albeit to various extent. As can be seen, the GroEL contamination is not the same in all samples. It ranges from ~5% to ~25% depending on the construct (as judged by visual inspection of the SDS-PAGE gel). Chaperone contamination is something we and others regularly observe when producing human proteins in a bacterial host. In all cases, though, the vast majority of the sample is our protein of interest. Therefore, we are not overly concerned about most of the protein not being folded properly. Otherwise, our filament structure could not have been deduced either, since it is obviously folded properly. And while the contamination with GroEL is not identical across the spectrum of mutants, none of them show filament formation, strongly arguing for this behavior not being a GroEL artifact. This may be most obvious

for the D188Y mutant, which has low GroEL contamination, comparable to wild-type, yet no filaments are observed.

The fact that we do have GroEL contamination in our samples readily explains why we can observe GroEL in the micrographs.

Electron-micrographs are not well-suited for quantification. For example, not oligomerized TorsinA will not be visible on the micrographs because of its small size, yet we know that it is ~30-50% of the sample already in wildtype (Suppl. Fig. 1c). Therefore, just seeing rings in a micrograph does not tell anything about the protein distribution in the sample, it only provides a qualitative measure. Furthermore, gel filtration of wildtype TorsinA does not indicate TorsinA ring formation since the elution profile does not show a peak at the expected molecular weight (at least 6×75 kDa = 450 kDa) (Suppl. Fig. 1c). Ultimately, we would have to do 2D classification of frozen grids to show that we only have heptameric GroEL particles, and no multimeric TorsinA rings. We believe that such an analysis goes beyond the scope of this paper.

Reviewer #2 (Remarks to the Author):

The authors have constructively addressed our concerns and the revised manuscript has improved considerably over the original version. These modifications also made the manuscript more accessible for a broader audience. We therefore recommend publication in the present form.

Reviewer #3 (Remarks to the Author):

I have no concerns with the revised manuscript, and support publication at this time.

Chris Hill

Reviewer #4 (Remarks to the Author):

The revised manuscript by Demircioglu and co-authors has been improved. Some answers however should be clarified.

Answer to the first reviewer:

The presented image is not convincing: the molecular images do not look like GroEL: side view should have four layers (or even more if it will be a complex with GroES) , the end views should have 7-fold symmetry. The presented views have apparently a 6-fold symmetry, but to be sure, the authors were supposed to do symmetry analysis of end views. From the presented figure it is rather difficult to assess sizes of the molecules, there are other smaller complexes. SDS or native gel will help to identify which species are existing in the sample. (Supl. Figure 5b). So the issue was not addressed.

What is shown on other panels of Supl. Figure 5? One can hardly see anything: particles are so small. The particle images in other panels have to be analyzed and class averages should be shown with sizes.

Response:
See comments above.

Answers to the fourth reviewer:

A2. The answer is not satisfactory; possibly a figure would help.

Response;
We have updated the introduction to restate the key differences to Clp proteins. Since this is not a review article we feel strongly that further elaborations about those differences are not warranted in this paper about Torsin filaments.

A6. Yes, it seems to prove the polarity of the TorsinA tubes at binding to membranes will be difficult. These experiments need a bit more time and thinking. However, the claim, that this is an unambiguous fact, that protrusions are covered by the protein, is extremely overrated. That was not proved by any experiments, which are not difficult to do.

Response:
We remain perplexed why the reviewer remains unconvinced that the lipid protrusions are protein covered, when all other reviewers seem to have no problem with this very straightforward and logical explanation. We agree that we have not formally proven this, i.e. by an antibody staining experiment, yet we think that our logical argument should be convincing enough without further proof.

A7. Thanks to the authors, they have improved the figures and have shown that a resolution is around 5A.

A8. This diffraction pattern is still very confusing; it is advisable to remove it from the MS: the figure is not informative and misleading. The authors have many other good and interesting figures.

Response:
We completely fail to see how the power spectrum shown in Supp. Fig. 2 is either confusing or misleading, and this was fully explained in our previous Response. Similar presentations of power spectra to what is shown in Supp. Fig. 2 are seen if one looks at recent papers from Egelman, such as:

Wang, F., Burrage, A.M., Postel, S., Clark, R.E., Orlova, A., Sundberg, E.J., Kearns, D.B. and Egelman, E.H. (2017). “A Structural Model of Flagellar Filament Switching Across Multiple Bacterial Species”, *Nature Communications* **8**, 960.

Spaulding, C.N., Schreiber, H.L.t., Zheng, W., Dodson, K.W., Hazen, J.E., Conover, M.S., Wang, F., Svenmarker, P., Luna-Rico, A., Francetic, O., Andersson, M., Hultgren, S. and Egelman, E.H. (2018). “Functional role of the type 1 pilus rod structure in mediating host-pathogen interactions.” *eLife* 7:e31662.

Wang, F., Gu, Y., O’Brien, J.P., Yi, S.M., Yalcin, S.E., Srikanth, V., Shen, C., Vu, D., Ing, N.L., Hochbaum, A.I.*, Egelman, E.H.* and Malvankar, N.S.* (*=corresponding authors) (2019). Structure of Microbial Nanowires Reveals Stacked Hemes that Transport Electrons over Micrometers. *Cell* 177, 361-369

Wang, F., Cvirkaite-Krupovic, V., Kreutzberger, M.A.B., Su, Z., de Oliveira, G.A.P., Osinski, T., Sherman, N., DiMaio, F., Wall, J.S., Prangishvili, D., Krupovic, M. and Egelman, E.H. (2019). “An extensively glycosylated archaeal pilus survives extreme conditions.” *Nature Microbiology*, in press (available online).

The coherent result of the reconstruction has to show the improved diffraction pattern that should be comparable with the diffraction pattern from the TorsinA tube. These diffractions are shown. For comparison see papers by N. Unwin; the recent one is in IUCrJ, 2017, V 4(4):393-399, <https://doi.org/10.1107/S2052252517005243> .

A9. It is an interesting hypothesis that lipids are located inside of the tube. If the authors are so sure of that the protein is located on outer surface of protrusions, the fab labeling will be rather easy and will remove any ambiguities. The author’s answer does not explain why the protrusions have roundish ends and smooth curved transitions between the liposome and the protrusions. It seems that lipids are located on the outer surface, then the model shown in Figure 6 is not correct. It would be sad if the respectful authors will make a mistake. May be lipids sit between protein subunits?

Response:

Based on steric arguments we remain utterly unconvinced that TorsinA should ever be on the inside of the membrane protrusions, as the reviewer seems to imply. Given the membrane topology of the starting material – liposomes – and the conditions used we cannot see how TorsinA should be on the inside of the protrusions. Again, it seems to be a peculiar concern of this reviewer, that apparently is not an issue for any other reviewer.

Comments: “Our in vitro experiments performed with liposomes suggest that the inner channel of these filaments may bind directly to lipids, likely relevant to elucidating the

enigmatic function of TorsinA.”

The correction of the previous version of the sentence should still be improved. The authors have to indicate which parts of the tube wall may bind lipids. The channel does not bind anything by itself.

Response:

The relevant sentence in the text has been adjusted.